# Metal-Ion Intercalation Mechanisms in Vanadium Pentoxide and Its New Perspectives

**DOI:** 10.3390/nano13243149

**Published:** 2023-12-15

**Authors:** Ricardo Alcántara, Pedro Lavela, Kristina Edström, Maximilian Fichtner, Top Khac Le, Christina Floraki, Dimitris Aivaliotis, Dimitra Vernardou

**Affiliations:** 1Departamento de Química Inorgánica e Ingeniería Química, Instituto Químico para la Energía y el Medioambiente (IQEMA), Universidad de Córdoba, Campus de Rabanales, Edificio Marie Curie, E-14071 Córdoba, Spain; iq1lacap@uco.es; 2Department of Chemistry—Ångström Laboratory, Uppsala University, SE-751 21 Uppsala, Sweden; kristina.edstrom@kemi.uu.se; 3Institute of Nanotechnology, Karlsruhe Institute of Technology, Hermann-von-Helmholtz-Platz 1, 76344 Eggenstein-Leopoldshafen, Germany; maximilian.fichtner@kit.edu; 4Helmholtz Institute Ulm (HIU) Electrochemical Energy Storage, Helmholtzstraße 11, 89081 Ulm, Germany; 5Faculty of Materials Science and Technology, University of Science, Ho Chi Minh City 700000, Vietnam; lekhactop@gmail.com; 6Vietnam National University, Ho Chi Minh City 700000, Vietnam; 7Department of Electrical and Computer Engineering, School of Engineering, Hellenic Mediterranean University, 71410 Heraklion, Greece; christinafloraki@hmu.gr (C.F.); d.aivaliotis92@gmail.com (D.A.); 8Institute of Emerging Technologies, Hellenic Mediterranean University Center, 71410 Heraklion, Greece

**Keywords:** vanadium pentoxide, univalent ions, multivalent ions, dual-ion intercalation, cationic doping, electrolytes, interfaces

## Abstract

The investigation into intercalation mechanisms in vanadium pentoxide has garnered significant attention within the realm of research, primarily propelled by its remarkable theoretical capacity for energy storage. This comprehensive review delves into the latest advancements that have enriched our understanding of these intricate mechanisms. Notwithstanding its exceptional storage capacity, the compound grapples with challenges arising from inherent structural instability. Researchers are actively exploring avenues for improving electrodes, with a focus on innovative structures and the meticulous fine-tuning of particle properties. Within the scope of this review, we engage in a detailed discussion on the mechanistic intricacies involved in ion intercalation within the framework of vanadium pentoxide. Additionally, we explore recent breakthroughs in understanding its intercalation properties, aiming to refine the material’s structure and morphology. These refinements are anticipated to pave the way for significantly enhanced performance in various energy storage applications.

## 1. Introduction

Vanadium pentoxide (V_2_O_5_) features a layered crystal structure, consisting of vanadium oxide layers separated by intercalation sites [1] (Figure 1). VO_5_ pyramids exhibit a remarkable degree of flexibility compared to other polyhedral structures, enabling more efficient support for reversible intercalation. In the case of orthorhombic V_2_O_5_, it demonstrates the capability to absorb water from both atmospheric conditions and solutions, resulting in the formation of V_2_O_5_·nH_2_O [2,3]. Notably, some of this water can be intercalated between the bilayers of V_2_O_5_. The absorbed water molecules manifest as neutral molecules (H_2_O) or oxonium cations (H_3_O^+^) due to the inherent acidity of V^5+^ ions. The presence of protons is often associated with acidic conditions (i.e., in acid–base equilibrium, protons may be involved in the transfer of H^+^), contributing to the negative charge within the V_2_O_5_^m−^ layers. This material serves as a protonic conductor, allowing for the exchange of water with organic solvents in nonaqueous electrolytes and the exchange of oxonium cations with outer cations (Li^+^, Mg^2+^, Al^3+^, etc.) in the electrolyte solution. Notably, the deintercalation of protons may coincide with the intercalation of other cations in battery applications, as highlighted in a study by González et al. [4]. The cationic exchange between vanadium oxide and the solution is particularly pronounced in amorphous or poorly crystallized vanadium oxides.

The layered structure of V_2_O_5_ facilitates the intercalation of various ions, including lithium and sodium [5,6,7], rendering vanadium oxide a viable choice as an electrode active material in lithium-ion batteries (LIBs). The success of LIBs as widely used power sources for portable electronic devices stems from their impressive energy density, allowing them to store a substantial amount of energy within a compact and lightweight structure. LIBs boast a low self-discharge rate, ensuring a prolonged retention of charge during periods of inactivity, thereby enhancing the reliability of electronic devices. Moreover, their extended cycle life contributes to long-term economic advantages. Beyond lithium, other elements such as sodium and magnesium, which are more abundant, pave the way for the emergence of “post-lithium batteries”, presenting potential competitiveness against LIBs in terms of cost and sustainability.

**Figure 1 nanomaterials-13-03149-f001:**
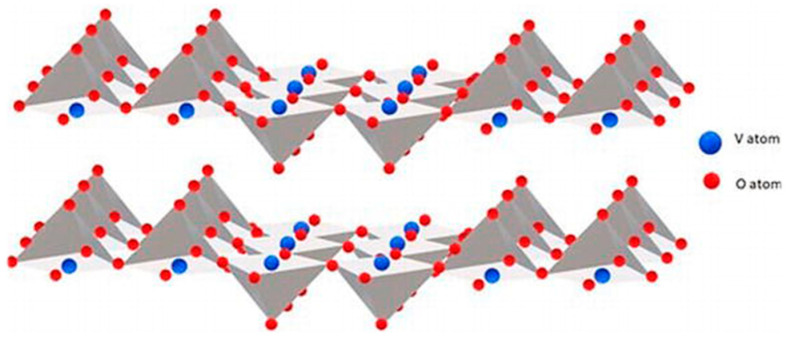
Perspective view of two V_2_O_5_ layers [8].

Vanadium pentoxide, known for its ease of fabrication, has sparked considerable interest in intercalation reactions, particularly within the realm of LIBs. Recently, there has been a growing consideration of post-lithium batteries, which could potentially rival lithium-based systems in terms of the natural abundance of elements, sustainability, and cost-effectiveness. The wealth of experience accumulated in the field of LIBs offers valuable insights that could pave the way for the development of innovative post-lithium battery technologies. This review aims to thoroughly investigate the application of V_2_O_5_ in batteries. The exploration begins by reviewing the methods of synthesis, and then examining the intercalation of both univalent and multivalent ions. The discussion finally delves into pivotal strategies designed to further enhance battery electrochemistry through its utilization.

## 2. Synthetic Methods

The morphological characteristics of electrode materials are significantly influenced by the chosen preparative routes, creating a notable impact on the nature of the electrode–electrolyte interphase and, consequently, the accessibility of insertion ions. It is paramount to carefully consider the selection of synthetic methods to obtain a balance between performance, cost, and environmental impact. This section aims to present some of the pertinent synthetic methods documented in the literature, underscoring the importance of each approach in achieving materials with optimized performance and balanced environmental considerations.

### 2.1. Sol–Gel

This method relies on dissolving chemical reagents to guarantee a comprehensive blending of elements. Frequently, soluble salts of the composing elements are dissolved in aqueous media, such as sodium vanadate and ammonium vanadate (NH_4_VO_3_), although vanadium alkoxides (e.g., vanadyl acetylacetonate) have been also often employed. To circumvent the selective precipitation of insoluble salts with varying compositions, a chelating agent is introduced before evaporating the solvent, such as an alkoxy ligand (e.g., triisopropoxide and butoxyacetylacetonate). This process results in the creation of a stable and uniform gel, ensuring a homogeneous mixture of elements at the atomic level. The intimate blending of these composing elements facilitates the production of highly crystalline samples characterized by a uniform particle distribution and nanometric size, even at low synthesis temperatures and short heating times [9].

During the last few decades, many gel products of V_2_O_5_ such as hydrogels, xerogels, aerogels, bronzes, and other oxides have been prepared by this method [10].

### 2.2. Hydrothermal Process

The hydrothermal process relies on the effects of high temperature and pressure to create an environment where the solvent, typically water, can dissolve reagents that are otherwise scarcely soluble. This facilitates the reaction process and leads to the precipitation of the desired product. Notably, this technique generally produces products that are highly pure and crystalline, with a straightforward operational procedure. However, it is important to acknowledge that the reaction yield tends to be low, potentially impacting its eventual industrial applicability.

To enhance the material’s crystallinity and consequently improve the electrochemical performance, a subsequent thermal treatment can be applied. The strategic use of appropriate surfactants can guide crystal growth in specific directions, enabling the preparation of samples with a controlled size and shape [11]. For instance, V_2_O_5_ nanotubes can be synthesized at 180 °C for 7 days with the addition of PEO as a surfactant [12]. The study’s findings indicated that these nanorod assemblies exhibit a lower specific capacity at initial stages but demonstrate superior cyclability compared to dispersed V_2_O_5_ nanotubes.

### 2.3. Spray-Drying

The spray-drying process aims to produce finely powdered granular products. This involves atomizing the precursor’s solution into suspended droplets, resulting in the formation of globular powders after rapid solvent evaporation in a hot gas stream. The maintenance of negative pressure in the drying room ensures the creation of pure products, while preventing the dispersion of airborne dust.

Lin et al. conducted the synthesis of V_2_O_5_ using ultrasonic spray pyrolysis at varying temperatures [13]. At 500 °C, they successfully produced spherical and densely nanostructured V_2_O_5_ particles, while observing a significant change in morphology as the synthesis temperature increased to 700 °C. The spherical V_2_O_5_ underwent testing, revealing an initial discharge capacity of approximately 403 mAh g^−1^. However, issues with retention arose due to inherent phase changes occurring at low voltages. Notably, adjusting the voltage limit from 1.5 to 2.0 V resulted in improved cycle performance for the V_2_O_5_ cathode. On a similar note, Ng et al. employed a one-step and scalable flame spray pyrolysis process to prepare V_2_O_5_ nanoparticles ranging from 30 to 60 nm [14]. Through careful optimization of the processing conditions, such as the precursor concentration and injection rate, they achieved particles with the lowest specific surface area and highest phase purity. This material demonstrated remarkable performance, retaining 110 mAh g^−1^ after 100 cycles at 100 mA g^−1^. Moreover, it exhibited a superior specific charge, reaching 100 mAh g^−1^ at 2000 mA g^−1^. Recent studies further corroborated that this method can be very convenient for the preparation of vanadium pentoxide electrodes [15].

### 2.4. Electrospinning

The electrospinning method involves a precursor solution within a capillary. To overcome surface tension and serve as a structural directing template for fiber formation, a polymer is introduced. Once the electrostatic field applied between the capillary and the receiving surface surpasses the surface tension of the droplet, a fine jet of precursor is ejected. Following solvent evaporation and solidification in the air, nanofibers are meticulously formed and collected. Despite being a straightforward and adaptable approach for crafting nanomaterials, the ultimate product’s characteristics are contingent upon specific attributes of the precursor solution. Factors such as the molecular weight of chemical reagents, stirring speed and duration, as well as environmental temperature and moisture play pivotal roles in determining the final outcome [16]. Common vanadium precursors are vanadium isopropoxide, vanadyl acetylacetonate, ammonium metavanadate, and V_2_O_5_ powder, while the polymers include poly(vinyl acetate), poly(methylmethacrylate), and poly(vinyl alcohol) [17]. Nevertheless, concerning issues such as low efficiency and the complexity of numerous process parameters along with the production of small material quantities pose a hurdle to achieving a widespread industrial applicability.

Both V_2_O_5_ fibers [18] and composite V_2_O_5_/C fibers [19] can be prepared. Initial investigations into the synthesis of V_2_O_5_ revealed that the morphology of the fibers could be precisely tailored by adjusting the annealing temperature. Furthermore, by adeptly controlling the parameters during calcination, the prospect of crafting V_2_O_3_ and VO_2_ nanofibers becomes a feasible consideration [18]. The fibers consisted of linked V_2_O_5_ particles. Tolosa et al. used a multi-step procedure including sol−gel synthesis, followed by electrospinning and controlled thermal treatment to prepare hybrid fibers composed of V_2_O_5_ and VO_2_ engulfed in conductive carbon. The former V_2_O_5_/C presented a high specific capacity of 316 mAh g^−1^, though the rate handling was less favorable [19].

### 2.5. Electrochemical Deposition

Electrochemical deposition relies on the reaction occurring at the surface of the electrode in an electrolytic cell. This process involves the application of a current, prompting the precipitation of insoluble products that coalesce into thin films. Achieving an optimal thin-film quality hinges on carefully managing factors such as the electrolyte concentration, pH, current density, cell potential, deposition time, and temperature [20,21,22].

Tepavcevic et al. reported nanostructured bilayered V_2_O_5_ prepared by the electrochemical deposition of 0.1 M VOSO_4_ aqueous solution on pure Ni foil, followed by vacuum annealing at 120 °C [23]. Sodium insertion involved the appearance of both long- and short-range order. Nevertheless, sodium extraction induced the loss of long-range order, whereas short-range order was preserved. This cathode material performed 250 mAh g^−1^ and a power density of 1200 W kg^−1^.

### 2.6. Template

This preparative method employs a template substance, with options including both hard and soft templates. Hard templates feature precisely defined void spaces, often comprised of channels, pores, or interconnected hollow areas. On the other hand, soft templates typically involve polymers and organic surfactants. In a standard synthesis process, the templating substance is blended or impregnated with the liquid precursor, leading to the formation of the solid product after a chemical reaction. Subsequently, the template must be removed to complete the process.

The wide array of structural and morphological possibilities inherent in various template precursors enables the creation of a diverse range of nanostructured electrode materials, encompassing one-dimensional nanowires or nanotubes, two-dimensional films, and three-dimensional interconnected porous architectures. Electrode materials crafted through this method typically showcase advantageous morphological traits, such as reduced particle size, expansive surface area, and robust structural integrity. These properties facilitate swift charge transfer and foster an optimal electrode–electrolyte interphase, playing a pivotal role in achieving superior electrochemical performance [24].

In the initial findings, hexadecylamine emerged as a soft template in structurally guiding the creation of vanadium oxide nanotubes, resulting in an impressive specific capacity of up to 180 mAh g^−1^. However, it is worth noting that the authors observed a gradual decrease in charge over successive cycles, attributed to a decline in electroactivity [25]. Recently, Youn et al. reported (1D) nanostructured VO_2_ (B) and V_2_O_5_·nH_2_O single-crystalline nanobelts prepared with cellulose nanocrystals (CNCs) as templating material. These nanobelts exhibited a high specific capacity (>300 mAh g^−1^) and long lifespan (>244 mAh g^−1^ at 50 cycles) [26].

## 3. Intercalation in Battery Electrode

### 3.1. Univalent Cations

#### 3.1.1. Lithium

The intercalation sites within V_2_O_5_ act as open spaces where lithium ions can insert themselves between the layers. During the charging process, these intercalation sites undergo reduction as lithium ions are inserted, causing the reduction from V^5+^ to V^4+^, or even V^3+^. As the lithium ions’ intercalation increases into the V_2_O_5_ cathode, the crystal structure of V_2_O_5_ expands to accommodate the inserted lithium ions. The resulting compound is typically represented as Li_x_V_2_O_5_, where “x” is the variable number of lithium ions that has been intercalated into the structure [27]. Depending on the extent of Li-ion intercalation (x), the structural evolution of orthorhombic V_2_O_5_ can undergo various phase transitions, leading to the formation of α-, ε-, δ-, and ω-phases. Figure 2 [28] indicates the crystal structures of various phases in V_2_O_5_. It is indicated that lithiation-induced volume expansion from α-V_2_O_5_ to δ-Li_x_V_2_O_5_ is near 11% [19]. The method of cyclic voltammetry (CV) is convincing to characterize the insertion number of the Li_x_V_2_O_5_ phases (Figure 2) [29]. In that perspective, the phase transitions are reflected in three distinct potentials at about 3.4 (Li_0.5_V_2_O_5_), 3.2 (LiV_2_O_5_), and 2.3 V (Li_2_V_2_O_5_) that correspond to α/ε, ε/δ, and δ/γ, respectively [19], with the following consecutive reactions [29] taking place:
V_2_O_5_ + 0.5Li^+^ + 0.5e^−^ ↔ Li_0.5_V_2_O_5_
Li_0.5_V_2_O_5_ + 0.5Li^+^ + 0.5e^−^ ↔ LiV_2_O_5_
LiV_2_O_5_ + 1Li^+^ + 1e^−^ ↔ Li_2_V_2_O_5_

The α-phase is present in Li_x_V_2_O_5_ when x < 0.1, while the ε-phase is observed within the range 0.35 < x < 0.7, resulting in a weak puckering of V_2_O_5_ [30,31]. When x is between 0.7 and 1, the δ-phase appears with a higher degree of puckering in the V_2_O_5_ layers [23]. Beyond x = 1, an irreversible transformation occurs to the γ-phase. The γ-phase can be reversibly cycled in the range 0 ≤ x ≤ 2 corresponding to theoretical capacity of 294 mAh g^−1^ [23]. Further intercalation of a third lithium ion results in the formation of the ω-phase, which has a tetragonal structure with lattice parameters of a = 4.1 Å and c = 9.2 [19].

During discharging, lithium ions begin to deintercalate from the cathode and move back into the anode and the lithium-ion electrolyte, involving the oxidation of V^4+^ or V^3+^ back to V^5+^. In Li_x_V_2_O_5_ with x ≤ 1, the formation of the α, ε, and δ phases is observed. The orthorhombic α-, ε-, and δ-phases consist of [VO_5_] square pyramids and exhibit significant puckering [32], while the γ-phase is irreversible and metastable (γ/γ’), transforming completely upon discharging. The ω-phase indicates irreversible structural changes and a rapid loss in capacity values with increased cycling.

Nevertheless, there is an irreversible transformation of γ-Li_x_V_2_O_5_ from δ-Li_x_V_2_O_5_ when discharge takes place beyond x = 1 as one can see from the structural relationship between the two phases [33]. In specific, there is a weak puckering of the double chains of VO_5_ pyramids in δ-V_2_O_5_, while it is more pronounced for γ-Li_x_V_2_O_5_, with the VO_5_ pyramids alternate up and down individually instead of forming pairs as in δ-Li_x_V_2_O_5_.

Knowledge of the kinetic parameters of Li^+^ insertion in correlation with the chemical diffusion coefficient is important to obtain information on the energy/power features of the lithium cells and a better understanding of the mechanism of lithium insertion in V_2_O_5_. A study conducted by Farcy et al. [34] through electrochemical impedance spectroscopy shed light on the influence of lithium content on the electroinsertion in V_2_O_5_, revealing discrepancies at high and intermediate frequencies with the absence of well-defined semicircles, as depicted in Figure 3. These deviations can be attributed to the porous nature of the electrode, leading to distinct potential and concentration distributions at the electrode–electrolyte interface [35]. In the spectrum, a discernible straight line emerges at lower frequencies, indicative of the Warburg impedance. The position of this impedance is contingent upon the lithium intercalation content denoted by the variable ‘x’ in Li_x_V_2_O_5_.

Exploring the diffusion coefficient of Li^+^ for different intercalation levels yields valuable insights [34]. During the initial stages of the intercalation processes, the rate of lithium diffusion experiences a significant reduction due to interactions between Li^+^ ions and the V_2_O_5_ host lattice. This phenomenon aligns with existing research, emphasizing that lithium intercalation induces structural modifications in V_2_O_5_, dependent on the composition of the bronzes [36]. Consequently, the diffusion coefficient reaches its maximum at lower lithium contents, gradually decreasing as ‘x’ increases. This diminishing trend is attributed to an escalation of the ionic interactions within the crystal lattice, constraining the mobility of the lithium ions. The maximum diffusion coefficient is observed at the lowest intercalation contents, decreasing approximately two orders of magnitude as ‘x’ varies from 10^−2^ to 0.1. Notably, a minimum value is observed for ‘x’ close to 0.25.

In a noteworthy study conducted by Światowska-Mrowiecka et al. [37], it was demonstrated through X-ray photoelectron spectroscopy (XPS) analysis that the process of lithium intercalation in V_2_O_5_ thin films, prepared by thermally oxidizing vanadium metal, is not entirely reversible. Specifically, after the first deintercalation step, 34% of the vanadium ions remained in the V^4+^ state, and after the second deintercalation step, this percentage decreased to 14%. This observation indicates that approximately 8% of the vanadium ions undergo irreversible reduction to V^4+^ following deintercalation. This phenomenon is attributed to the entrapment of a portion of the intercalated Li^+^ ions within the V_2_O_5_ films, likely occurring at the grain boundaries of the crystalline oxide structure.

#### 3.1.2. Sodium

In recent years, sodium-ion batteries (SIBs) have emerged as a viable alternative to LIBs due to several advantages. The abundance of sodium as a resource results in lower material costs for SIBs. It exhibits enhanced safety characteristics compared to lithium, as it is less reactive and less prone to thermal runaway events. Moreover, SIBs demonstrate similar voltage levels and charge/discharge profiles to LIBs, making them compatible with existing battery technologies. They follow a similar principle of operation as LIBs, utilizing sodium ions as the charge carriers. Although the investigation of SIBs began alongside LIBs in the 1970s–1980s [38], they witnessed a significant decline with the successful commercialization of LIBs in the 1990s. However, the growing need for improved battery performance has reignited interest in SIBs.

To enhance the performance of SIBs, particular attention has been given to cathode materials. These materials face challenges such as structure degradation, low capacity, and slow kinetic diffusion properties due to the larger size of sodium ions (1.02 Å for Na^+^) compared to lithium ions (0.76 Å for Li^+^) [39,40]. Therefore, research efforts are focused on exploring and developing cathode materials that can overcome these limitations and deliver improved performance in SIBs.

Orthorhombic V_2_O_5_ has been recognized as a promising cathode due to its open structure, which enables the reversible accommodation of Na^+^ ions. The crystal structure of layered V_2_O_5_, which has an orthorhombic *P_mmn_* lattice [41], undergoes a phase transition when sodiated [42]. This structure is composed of edge- and vertex-sharing VO_5_ square pyramids (Figure 4, left [43]) throughout the crystal structures of the bronzes, which are exceptionally flexible as exemplified with the layered (α-phase) and tunnel (β-phase) materials.

The sodium-vanadium bronzes, denoted by the general formula Na_x_V_2_O_5_ (0 < x ≤ 2), are a diverse family of compounds with mixed-valence states of vanadium ions ranging from V^4+^ to V^5+^. Unlike bronzes of other transition metals, this family encompasses a wide range of structures classified as α-, β-, γ-, δ-, τ-, α’, η-, κ-, and χ-phases [44,45,46,47,48]. The phase of Na_x_V_2_O_5_ depends on the value of x. At a low sodium content (0 < x < 0.2), it exists in the α-phase with no substantial alterations in the lattice parameters [38]. In the range of 0.2 < x < 0.4, the β-phase is formed [36], while the specific composition Na_0.64_V_2_O_5_ is called the τ-phase [32]. This transition may be associated with modifications in the arrangement of vanadium and sodium ions. In 1988, Jacobsen first conducted a study on α-V_2_O_5_, channel-type β-Na_x_V_2_O_5_, and layered Na_1+x_V_3_O_8_ in all-solid-state batteries using poly(ethylene oxide)-based electrolyte at 80 °C [49]. This material exhibited the insertion of 2 moles of Na^+^ during the initial discharge, followed by the reversible discharge of 1.7 moles of Na^+^ in subsequent cycles [41]. Monoclinic β-Na_0.33_V_2_O_5_ has also gained attention because of the tunnel structure, adopting three different Na intercalation sites and ensuring good structural reversibility even upon deep charge/discharge processes [33].

The α’-phase can be found when the sodium content is between 0.7 and 1 [32]. For higher sodium contents, the η-phase is observed in the range of 1.45 < x < 1.8, while the κ-phase occurs in the range of 1.68 < x < 1.82 [41]. The compound η-Na_x_V_2_O_5_ or Na_9_V_14_O_35_ crystallizes in a monoclinic lattice with the space group P2/c [50]. Its crystal structure consists of layers oriented along the (010) direction, which are composed of VO_5_ (V^4+^) square pyramids and VO_4_ (V^5+^) tetrahedrons with the sodium atoms interspersed between these layers (Figure 4, right) [43]. If all sodium ions are extracted from the structure, the theoretical capacity of η-Na_x_V_2_O_5_ can be estimated to be approximately 163 mAh g^−1^. This value, coupled with potential electrochemical activity in the anodic region, highlights the material’s potential as a new intercalation system for sodium ions. Understanding the conditions that lead to η-, κ-, and χ-phases can provide insights into the material’s behavior.

Electrochemical impedance spectroscopy is also utilized to investigate the Na^+^ kinetics reaction for a deep understanding of the sodium insertion–extraction mechanism in γ’-V_2_O_5_ [51]. γ’-V_2_O_5_ is characterized by a high (001) preferred orientation corresponding to the stacking of platelets along the c-axis. It exhibits a layered structure differing from that of α-V_2_O_5_, distinguished by a notable layer puckering. This distinctive feature involves the assembly of infinite ribbons, comprised of VO_5_ edges-sharing distorted pyramids and accentuated by a generous interlayer spacing of 5.02 Å (in contrast to the 4.37 Å observed in α-V_2_O_5_). These ribbons interconnect along the a-direction, giving rise to intricately puckered slabs that align in stacks along the c-direction, as illustrated in Figure 5.

The crucial electrochemical parameters of the sodiated electrode such as cathode impedance, charge transfer resistance, and double-layer capacity are indicated in Figure 5. The cathode impedance exhibits low values, typically ranging from 300–400 Ω, except for instances of very low sodium content, indicative of high cathode impedance. As sodium uptake progresses, the cathode impedance follows suit, rising from approximately 100 Ω at x = 0.2 to 350 Ω at x = 0.6 and peaking at 700 Ω for x = 0.97 (Figure 5a). Notably, the fully sodiated sample displays a cathode impedance nearly double that of a sodium-free electrode, underscoring the challenging nature of sodium extraction. Concurrently, the gradual and linear growth of R_ct_ with x (as illustrated in Figure 5b) signifies a deceleration in charge transfer kinetics during discharge. In the biphasic region (where γ’-V_2_O_5_ and γ-Na_0.7_V_2_O_5_ coexist), R_ct_ increases from 22 to 38 Ω, while a more pronounced increase (by a factor of 2) is observed in the 0.7 < x ≤ 0.97 solid-solution domain. This trend highlights the adverse impact of deep sodiation on electron transport, leading to a linear decrease in exchange current density with x, dropping from 1 mA cm^−2^ for γ’-V_2_O_5_ to only 0.4 mA cm^−2^ for γ-Na_0.97_V_2_O_5_. Turning attention to Figure 5c, the evolution of the double-layer capacity reveals compelling insights. The C_dl_ parameter demonstrates an increase from approximately 18 μF cm^−2^ for γ’-V_2_O_5_ to about 30 μF cm^−2^ for Na_0.97_V_2_O_5_. Notably, this substantial 50% increase in C_dl_ during Na insertion contrasts with the more modest 15% reported increase during Li insertion in γ’-V_2_O_5_. This discrepancy is attributed to the larger volume change induced by sodiation in γ’-V_2_O_5_ compared to that induced by lithiation (17% vs. 4.4%). Furthermore, examples from other studies underscore the close correlation between C_dl_ and volume change in cathode materials, such as a 10% increase in C_dl_ for a 5% volume change in nanosized V_2_O_5_ [52] and a constant C_dl_ for the Cr_0.11_V_2_O_5.16_ mixed oxide with a limited volume change not exceeding 3.5% [53].

The results show a strong correlation between the evolution of cathode impedance, charge transfer resistance, double-layer capacity, and the apparent chemical sodium diffusion coefficient with the structural changes induced during sodiation. In the region of the richest Na uptake (0.6 < x ≤ 0.97) in γ-Na_x_V_2_O_5_, a faster sodium diffusion is observed, while electron transport is slowed down due to the highly localized electron character of the sodiated phase. Other issues that need investigation is the relationship between the depth of discharge and the kinetics to provide insights of cycling performance optimizations along with strategies to accommodate volume changes and enhance the material’s ability to withstand the expansion and contraction cycles.

The Na^+^ ion intercalation/deintercalation reaction mechanism in V_2_O_5_ has also been investigated by X-ray diffraction (XRD), transmission electron microscopy (TEM), and near-edge X-ray absorption fine structure (NEXAFS) spectroscopy to gain a better understanding of the mechanisms involved and design more efficient electrode materials [54]. The changes in the lattice structure during Na^+^ intercalation/deintercalation and the charge–discharge process are summarized in Figure 6. Sodium intercalation and deintercalation occur along the ab planes with ordered chains of layers connected through corner and edge sharing. The intercalation of Na ions leads to an expansion of the interlayer distance along the c-axis, from 4.4 to 4.8 Å at the fully discharged rate. In that process, V_2_O_5_ undergoes a transformation to the crystalline phase of NaV_2_O_5_, while partially amorphous Na_2_V_2_O_5_ coexists as a minor phase. The Na^+^ ions in the NaV_2_O_5_ structure occupy the Wyckoff position of *2b* between the van der Waals, causing an expansion of the c-axis. During the charging process, Na^+^ ions are extracted, and V_2_O_5_ returns to its original crystal structure, with NaV_2_O_5_ present as a minor phase.

The electrochemical properties of α-V_2_O_5_ are investigated at room temperature for the first time in a 1 M NaClO_4_/PC (propylene carbonate) electrolyte. When Na^+^ ions are inserted into V_2_O_5_, the pristine α-V_2_O_5_ structure remained unchanged, with no significant alterations in lattice parameters observed at low sodium concentrations (x ≤ 0.2 in Na_x_V_2_O_5_) [55]. However, for sodium amounts in the range of 0.2 ≤ x ≤ 1.6, the insertion of Na ions leads to notable changes in the lattice parameters within 0.2 ≤ x ≤ 0.7, followed by slight changes in 0.7 ≤ x ≤ 1.6. During the first Na^+^ insertion, an irreversible phase called NaV_2_O_5_ is formed. However, this NaV_2_O_5_ phase can subsequently reversibly accommodate 0.8 Na^+^ within the potential range of 1.4 V to 3.0 V versus Na^+^/Na. To that point, it is worth mentioning that Ali and coworkers proposed an alternative reaction mechanism, suggesting the presence of a two-phase mixture consisting of NaV_2_O_5_ (major phase) and Na_2_V_2_O_5_ (minor phase) during the discharge process [44]. The Na_2_V_2_O_5_ includes crystalline and amorphous-like phases. Upon the charging process, the material returns to the original V_2_O_5_ structure with the presence of NaV_2_O_5_ minor phases.

Despite the existence of studies suggesting alternative reaction mechanisms and the observation of phase changes during the insertion and extraction of Na^+^ ions, a comprehensive understanding of the intricate processes involved in the Na^+^ insertion mechanism remains elusive. To fully elucidate the precise mechanism and shed light on the underlying processes occurring during Na^+^ insertion into V_2_O_5_, further research and investigation are required.

#### 3.1.3. Potassium

Potassium-ion batteries (PIBs) have gained attention due to the abundance (i.e., 17,000 ppm in Earth’s crust as compared with Li resource being only 20 ppm) and cost-effectiveness of potassium. The redox potential of K^+^/K (−2.93 V vs. standard hydrogen electrode (SHE)) is very close to that of Li^+^/Li (−3.04 V vs. SHE), indicating that PIBs and LIBs may have similar energy densities, theoretically [56].

PIBs have a similar working principle of “rocking star” with LIBs. However, they exhibit a lower performance compared to LIBs. The structural transformation from α-V_2_O_5_ to KVO [57,58] (Figure 7 [59]) involves the insertion of K^+^ ions into the layers, resulting in layer slippage. Notably, the distortion at the edges of the lattice enhances the probability of insertion occurring between the two layers. As a result, the VO_5_ square pyramid undergoes a backward rotation, following the typical motion observed in VO_5_. Driven by lower thermodynamic energy, the vanadium atoms within the lattice reorganize themselves and establish stable VO_6_ octahedra by forming new bonds with oxygen atoms. Ultimately, through this process of structural evolution, the KVO layer structure is formed.

The presence of a compact interlayer space and tightly arranged VO_5_ square pyramids in α-V_2_O_5_ poses difficulties for the insertion and diffusion of K^+^ ions within its crystal structure [60,61]. Theoretical strategies propose that lowering the oxidation state of vanadium in α-V_2_O_5_ could induce the transformation of square pyramids into octahedra, which would expand the crystal structure and create a larger interlayer space suitable for K^+^ ion diffusion [49]. However, achieving precise control over the oxidation states of vanadium presents challenges and may result in impurities within the material.

Additionally, the ionic radius of K^+^ (1.38 Å) compared with that of Li^+^ (0.76 Å) or Na^+^ (1.02 Å) limits the number of suitable host materials capable of accommodating K without large stresses or volume changes [29]. In addition, repulsive K^+^–K^+^ coulombic interactions could limit transport properties in the crystal framework and, as a consequence, the rate capability of the battery [62].

The potassium intercalation potential of the α-V_2_O_5_ polymorph has been relatively unexplored. Previous research has demonstrated the formation of an orthorhombic K_x_V_2_O_5_ bronze structure at high temperatures using a potassium electrolyte [63]. However, the extraction of potassium has been limited to only 50%. Recent investigations have focused on studying metastable V_2_O_5_ polymorphs that are obtained by removing cations from M_x_V_2_O_5_ bronzes through topochemical processes [64]. Density functional theory (DFT) calculations have revealed that these cation-free metastable phases exhibit lower diffusion barriers and higher operating voltages, which renders them promising for potassium storage [53].

The process of K^+^ ion insertion into α-V_2_O_5_ nanoparticles was thoroughly investigated using advanced techniques such as in operando synchrotron diffraction, in operando XAS, in situ Raman, XPS, and TEM [65]. The experimental observations revealed that during the initial stages of K^+^ ion insertion, V_2_O_5_ undergoes a solid-solution phase transformation, leading to the formation of the K_0.6_V_2_O_5_ phase. As the insertion process progresses, a combination of solid-solution and two-phase reactions occurs simultaneously. This suggests that the system deviates significantly from equilibrium, which can be attributed to the presence of kinetic barriers and transport limitations. Furthermore, the formation and subsequent degradation of a solid electrolyte interphase (SEI) were observed during the K^+^ ion insertion process. This indicates the occurrence of substantial side reactions, which have a distinct impact on the degradation mechanism of the material.

The γ’-V_2_O_5_ polymorph, which is derived from the γ-LiV_2_O_5_ phase through the electrochemical lithiation of α-V_2_O_5_, possesses advantageous characteristics. While maintaining orthorhombic symmetry like α-V_2_O_5_, γ’-V_2_O_5_ exhibits a larger interlayer spacing and a corrugated structure [62]. Interestingly, γ’-V_2_O_5_ can effectively accommodate a significant quantity of potassium (up to 0.9 K^+^ ions per mole) between its puckered layers at a voltage of 3.3 V versus K^+^/K (Figure 8) [62]. Detailed structural analysis within the voltage range of 4.4 V to 2.4 V has revealed that the potassiation process in γ’-V_2_O_5_ triggers a distinctive structural rearrangement, resulting in the crystallization of K_0.78_V_2_O_5_ in the P_nma_ space group. This transition is accompanied by a moderate interlayer expansion of 9.4% upon potassium insertion (in comparison to 18% during sodiation) [62]. In this new structure, the V_2_O_5_ layers consist of alternating up- and down-oriented VO_5_ pairs within the interlayer spacing, in contrast to the alternating single VO_5_ pyramids observed in γ’-V_2_O_5_. This change in structure highlights the unique capacity of the pristine γ’-V_2_O_5_ lattice to minimize structural modifications during the incorporation of K^+^ ions [62].

Bilayered vanadium oxide (δ-V_2_O_5_) represents another highly promising cathode material suitable for high-capacity applications [66]. Its structure comprises double layers of V-O, creating a notable interlayer spacing of approximately 11.5 Å, which is facilitated by the presence of water molecules [67]. This expanded interlayer space offers ample room for the intercalation of ions. By utilizing potassium ions (K^+^), both singly and doubly charged ions can be inserted within the interlayer region of the δ-V_2_O_5_ phase [68]. The resulting compound, known as δ-K_x_V_2_O_5_·nH_2_O, exhibits a bilayered structure with an interlayer spacing reduced to 9.62 Å [61]. Remarkably, it demonstrates an initial capacity of 226 mAh g^−1^ when employing an organic electrolyte [59]. The charge storage mechanism within the K-preintercalated δ-V_2_O_5_ phase is primarily governed by diffusion-limited intercalation [59].

It is important to highlight that although the intercalation mechanism of K^+^ ions in V_2_O_5_ exhibits similarities to Li^+^ ion intercalation, there are notable differences. The larger size of K^+^ ions compared to Li^+^ ions can introduce structural distortions and impose limitations on the intercalation process, resulting in lower performance when compared with LIBs. Furthermore, it is crucial to consider that the specific properties and performance characteristics can vary depending on the form of V_2_O_5_ employed.

### 3.2. Divalent Cations

#### 3.2.1. Magnesium

The intercalation mechanism of Mg^2+^ in V_2_O_5_ involves a reversible electrochemical process during charging/discharging according to the following reaction [69]:xMg^2+^ + 2xe^−^ + V_2_O_5_ ↔ Mg_x_V_2_O_5_

During the charging (insertion), Mg^2+^ ions are introduced into the structure of V_2_O_5_ through the open channels that exist between the layers of vanadium oxide. An oxidation-reduction (redox) reaction occurs, causing the Mg^2+^ ions to undergo oxidation and release electrons, which are then transferred to the electrode material. Consequently, the interlayer spacing between the vanadium oxide layers expands, enabling the incorporation of the magnesium ions within the V_2_O_5_ structure. The Mg^2+^ ions are subsequently stored within the V_2_O_5_ lattice, leading to the formation of a compound known as magnesium vanadium pentoxide (Mg_x_V_2_O_5_), where the variable x represents the extent of magnesium intercalation. When discharging takes place, the previously stored magnesium ions are released from the V_2_O_5_ lattice, moving in the opposite direction.

In the initial investigations conducted by Novák et al. [69], it was observed that the intercalation of Mg^2+^ into V_2_O_5_ is a slow and incomplete process at room temperature. The α-V_2_O_5_ phase is the most thermodynamically stable and is composed of double-chain sheets formed by square pyramidal units, which lie parallel to the α-axis (Figure 9) [70]. Upon the Mg^2+^ insertion, the oxidation of vanadium ions decreases from V^5+^ to V^4+^ and they become coordinated between the sheets of the V_2_O_5_ pyramids. In the α-phase, the calculated energy barrier for the insertion of Mg^2+^ is approximately 975 mV, causing irreversible Mg^2+^ processes. Additionally, it needs to be noted that the insertion of Mg^2+^ being higher than 0.5 mole is not practical since V_2_O_5_ undergoes a structural transition, transforming into the ε phase (Figure 9) [70], which has lower potential and a weak driving force for Mg^2+^ intercalation. Figure 9 indicates the ε phase corresponding to a specific ordering of Mg atoms in α-V_2_O_5_ at half magnesiation.

The realization of a functional magnesium-ion battery (MIB) with a high energy density remains an ongoing challenge. Consequently, layered V_2_O_5_ has been both the focus of such endeavors and a subject of controversy. More recent studies have indicated that α-V_2_O_5_ is capable of reversible intercalation of at least 1 mole of Mg^2+^ at 110 °C when using an ionic liquid electrolyte with high chemical and anodic stability [71]. Figure 10 indicates that α-V_2_O_5_ has a small reversible capacity of 16 mAh g^−1^ at 25 °C, which increased 20-fold to 295 mAh g^−1^ at 110 °C. This process resulted in the highest observed Mg capacities to date [72]. In particular, the ability to reversibly intercalate at least one Mg^2+^ per unit formula was demonstrated without significant co-intercalation of H_2_O or H^+^, nor competition with the conversion to MgO and VO_y_, for y < 2.5. The transformations occurring are related with two reaction mechanisms including: a. intercalation, where Mg^2+^ ions are inserted into the V_2_O_5_ structure, and b. conversion, involving the transformation of V_2_O_5_ into different compounds as a result of the Mg^2+^ insertion process [65].
V_2_O_5_ + xMg^2+^ + 2xe^−^ → Mg_x_V_2_O_5_
V_2_O_5_ +xMg^2+^ + 2xe^−^ → xMgO + VO_2.5−x_

By incorporating PEDOT, the interlayer spacing of V_2_O_5_ was controlled, resulting in the development of a bilayered V_2_O_5_/PEDOT (VOP) composite with an intercalation topology [73]. This composite demonstrated rapid and reversible charge storage for MIB cathodes. To gain insights into the mechanism of Mg^2+^ ion storage within VOP during the charge/discharge processes, in situ XRD, ex situ XRD, and ex situ XPS analyses were performed.

During the insertion and extraction of Mg^2+^ ions between the interlayers of VOP, the Mg^2+^ ions undergo solvation by water molecules promoting the activation process and lowering the desolvation energy [73]. The interlayer spacing of the (001) plane of VOP was measured to be 19.02 Å in the charged state at +1.0 V (Figure 11, left), which is approximately 4.37 times larger than that of pristine V_2_O_5_. In addition, the interlayer spacing in the discharge state at −1.0 V was estimated to be 20.16 Å (Figure 11, right) [73]. These measurements provide valuable information about the structural changes occurring within VOP during the intercalation and deintercalation of Mg^2+^ ions.

#### 3.2.2. Calcium

In contrast with the vast knowledge available about the Li^+^ intercalation mechanism, the nature and crystal structure of the phase formed for the case of Ca^2+^ are even less explored. The available DFT studies predict migration barriers of about 1.7–1.9 eV in α-V_2_O_5_ with an average voltage of about 3.2 V vs. Ca^2+^/Ca [71]. The polymorph α-CaV_2_O_5_ [71] prepared by solid-state reaction enables an appropriate coordination for the Ca^2+^ [74].

Seung-Tae Hong et al. [75] introduced a novel intercalation mechanism for Ca_0.28_V_2_O_5_·H_2_O wherein upon charging, Ca^2+^ ions are extracted from every two interlayers, leaving only a fraction of calcium ions in the remaining interlayers. Upon discharging, irregular insertions of Ca^2+^ ions occur within the interlayers, leading to stacking faults.

Another study presented the utilization of the acetonitrile–water hybrid electrolyte in Ca_0.25_V_2_O_5_·nH_2_O, referred to as CVO. The crystal structure of CVO, illustrated in the accompanying Figure 12 [76], reveals a layered configuration with inserted Ca^2+^ ions occupying the interlayer spaces of V_2_O_5_. These divalent ions form robust ionic bonds with oxygen atoms, creating polyhedral pillars (CaO_7_). These pillars effectively stabilize the V_2_O_5_ layers and prevent potential structural failure during the cycling of the battery [77].

Double-sheet V_2_O_5_·nH_2_O (Figure 13) was reported as a reversible host structure for Ca^2+^ ions [78]. The presence of water molecules (n) in the structure depends on synthetic and environmental factors. The crystal structure of double-sheet V_2_O_5_·nH_2_O consists of bilayers, with each bilayer composed of two opposing sheets of V_2_O_5_ [79]. These sheets are constructed using VO_5_ square pyramidal units, which are either corner- or edge-shared. The two V_2_O_5_ sheets face each other at a distance of approximately 2.8 Å, while the interlayer spacing between the double-sheet layers is greater than 8 Å, providing room for crystal water [72].

The mechanism for storing Ca^2+^ ions involves both bulk intercalation and surface capacitance reactions [70]. Analytical techniques such as XRD, TEM, elemental analyses, and bond-valence-sum map (BVSM) calculations were employed to investigate the Ca^2+^ transport and the electrochemical reaction. The results unequivocally indicate that the transportation of Ca^2+^ ions plays a significant role in the electrochemical reaction demonstrating a new type of calcium-intercalating host material [70].

### 3.3. Trivalent Cations

The chemical intercalation of aluminum in aerogel V_2_O_5_ with trimethylaluminum in heptane was reported in 1998 [80]. These chemically intercalated aluminum ions in Al_1.33_V_2_O_5_ were employed in an electrochemical cell for intercalation into a V_2_O_5_ counter-electrode, in a sort of Al-ion battery, using Al(CF_3_SO_3_)_3_ in propylene carbonate as the electrolyte solution.

Several authors have reported the intercalation of aluminum ions into V_2_O_5_ using AlCl_3_ aqueous electrolytes, although it is still debated. The deintercalation of aluminum in V_2_O_5_ using an aqueous solution takes place in the voltage range between ca. −0.3 V and +0.1 V vs. MSE with a reversible capacity of about 120 mAh g^−1^ (Figure 14, left) [81]. The low capacity at high current density is indicative of an intercalation reaction controlled by solid-state diffusion. The aluminum ion is highly polarizing, and its charge is shielded by the dipoles of the water molecules in its coordination sphere. The electrochemical reaction for the intercalation of hydrated aluminum in V_2_O_5_ can be written as
x/3Al(H_2_O)_n_^3+^ + (V^5+^)_2_O_5_ + xe^−^ ↔ (Al^3+^)_x/3_[(V^4+^)_x_,(V^5+^)_2−x_]O_5_·nH_2_O

The resulting Al-intercalated vanadium oxide is amorphous at approximately x = 0.5, in contrast to the crystalline character of the Li-intercalated compound. It is worth noting that the electrochemical behavior in an aqueous solution of aluminum could be better than that in an aqueous lithium solution, and this fact can be related to the great polarizing power of the aluminum ion and its tendency to be in hydrated form. However, the diffusion of the hydrated aluminum ion in the solid electrode can be sluggish. After preparing V_2_O_5_ particles with nanosheets-like morphology by hydrothermal method, the capacity is increased up to 140 mAh g^−1^ at high current density, and the capacity retention is excellent [82]. On the other hand, the interlayer spacing of V_2_O_5_ during the deintercalation of aluminum remains nearly unchanged.

Beyond aqueous electrolytes, binder-free V_2_O_5_ and an ionic liquid as an electrolyte also support the intercalation of Al^3+^ at around between 0.0 and 1.2 V vs. Al^3+^/Al [83]. Although most Al in the ionic liquid exists in the form of AlCl_4_^−^ and Al_2_Cl_7_^−^, it seems that the cation Al^3+^ is intercalated. The voltages of the peaks in the CV results suggest that aluminum prefers to intercalate on the inner layer sites (site-a) (Figure 14, right) [83].

### 3.4. Dual-Ion Intercalation

The hydrated vanadium oxide V_2_O_5_·nH_2_O, referred to as VOH, was utilized for the simultaneous intercalation of one-valent metal K^+^ and a divalent alkaline earth metal Mg^2+^, denoted as KMgVOH (Figure 15) [84]. The figure presents the structural models of the three samples, MgVOH, KVOH, and KMgVOH. Introducing Mg^2+^ or K^+^ into the V–O interlayer causes a change in the interlayer spacing. In specific, upon Mg^2+^ insertion, the spacing expands to 14.1 Å, while K^+^ intercalation leads to a spacing of 10.4 Å [84]. On the other hand, when both metal ions are simultaneously intercalated, the spacing remains the same as with VOH (i.e., 11.4 Å) [84]. Overall, it was observed that Mg^2+^ insertion increases the layer spacing of VOH, thereby widening the ion transport channels and enhancing the battery’s capacity [84]. Conversely, K^+^ intercalation brings the V–O layers closer together, leading to a stabilized structure for the material [84].

A synergistic mechanism involving the combination of Zn^2+^ and H^+^ (Figure 16) was successfully achieved in Li_0.45_V_2_O_5_·0.89H_2_O (LVO) nanoplates within 2 M ZnSO_4_ [85]. During the initial stage (1.5 to 0.7 V), both Zn^2+^ and a small number of H^+^ ions are inserted into LVO nanoplates, resulting in a continuous reduction of the interlaminar spacing from its original value of 10.45 Å to 9.45 Å [85]. This reduction is primarily attributed to the strong electrostatic attraction between the negatively charged VO_x_ layers and the positively charged divalent Zn^2+^ ions. As the interlayer distance decreases and active sites between the LVO layers become available for Zn^2+^ insertion, a second discharge stage is initiated (0.7 to 0.3 V), during which only H^+^ ions are inserted into the LVO with the formation of flake-like Zn_4_SO_4_(OH)_6_·0.5H_2_O (ZHS) structures on the surface of the LVO cathode [85]. In this second discharge stage, any excess of OH^-^ ions is consumed through the generation of these flake-like ZHS structures [85]. Finally, during the recharge process, the extracted H^+^ ions can dissolve the ZHS structures, which is advantageous for improving the cycle stability of the battery system.

## 4. Recent Advances in the Intercalation Properties of Vanadium Pentoxides

Enhancing electrodes derived from vanadium compounds can be achieved through two primary avenues: the exploration of novel structures and the fine-tuning of particles. As an alternative to low-pressure α-V_2_O_5_, the high-pressure variant of vanadium oxide, β-V_2_O_5_, has demonstrated potential viability as an electrode for both lithium and sodium cells [86,87]. Nevertheless, the synthesis procedure under high-pressure conditions appears to pose challenges for scalability in large-scale applications.

Reducing the particle size through ball-milling significantly enhances the electrochemical properties of γ′-V_2_O_5_, showcasing improved diffusivity and accelerated sodium diffusion within the single-phase region [88].

The integration of nanosheets from bilayered V_2_O_5_ with graphene layers to create a two-dimensional heterostructure holds the potential to expedite interfacial charge transfer, facilitating swift pseudocapacitive multi-electron reactions [89]. Through the strategic engineering of the vanadium oxide heterostructures, this approach aims to achieve high-energy and high-power lithium storage. Notably, the predominant contribution to charge storage in lithium cells for this V_2_O_5_/graphene heterostructure arises from pseudocapacitance.

To further refine the structure and morphology of V_2_O_5_, a recent synthetic method has been reported [90]. This involves the calcination of self-assembled surfactant 1,3-bis[(3-octadecyl-1)-imidazolio)methyl]benzene (gemini) and [V_10_O_28_]^6−^ clusters, resulting in orthorhombic V_2_O_5_ in the form of rod-like particles. The length of the rods increases with the calcination temperature.

### 4.1. Cationic Doping

The utilization of cationic doping in V_2_O_5_ cathodes is employed to enhance their electrochemical performance, addressing drawbacks such as their limited electronic and ionic conductivity as well as stability issues. By introducing extra charge carriers into the lattice, cationic doping facilitates the movement of electrons and ions during charge/discharge processes. This, in turn, improves conductivity and prevents structural deterioration throughout cycling [91,92,93,94,95]. Additionally, the incorporation of cationic doping serves to enhance the number of active sites available for ion intercalation. This, in turn, contributes to heightened ion storage capacity, translating into increased energy density and an extended lifespan for the battery.

Currently, there is no consensus among researchers regarding the optimal theoretical approach for treating doped V_2_O_5_. In an effort to address this issue, Jovanović et al. employed density functional theory (DFT) calculations to provide a general overview of the effects of 3d element doping (from Sc to Zn) in V_2_O_5_ [91]. Their study revealed that interstitial doping resulted in an expansion of the V_2_O_5_ lattice, while substitutional doping had a relatively minor impact on the structure [94]. However, it had a significant effect on the electronic structure, leading to a reduction in the band gap of V_2_O_5_ and an improvement in conductivity. This theoretical analysis confirms doping as a promising strategy for modifying the structural and electronic properties of V_2_O_5_.

Starting with Mn and Ni, it was indicated that the high concentration of both dopants in V_2_O_5_·nH_2_O clearly presented effects in morphology resulting in the presence of radial aggregates (Figure 17) [92]. The pre-intercalation of Ni and Mn in the interlayer space resulted in an improvement in the delivered capacity and rate capability, likely attributed to the presence of dopants and induced morphological changes. Notably, Ni-doped V_2_O_5_ electrodes exhibited a higher initial capacity (350 mAh g^−1^) compared to Mn-doped ones (290 mAh g^−1^). Overall, all the doped samples delivered higher specific capacities than the undoped ones. In the case of the doped samples, the irreversible capacity loss during the first cycle most likely originated by the Li^+^ reaction with the trapped water molecules in the V_2_O_5_ interlayer (Figure 18) [92].

In another study, the potential of Cu doping was investigated, as Cu is known for its advantageous combination of electrical conductivity and cost, making it a practical candidate for doping [96]. The researchers examined the electrodeposited V_2_O_5_ doped with 1 wt.% Cu, showing a discharge capacity of up to 362 mAh g^−1^, which corresponds to 86.2% of the theoretical capacity (420 mAh g^−1^) [97] for the insertion of two Li^+^ ions [98]. The enhanced capacity of the Cu-doped thin-film electrode, compared to the pure electrode, can be attributed to three factors: an increase in active area, the vine fiber structure, and the higher electronic conductivity resulting from Cu doping. The vine fiber structure provides more sites for the embedding Li^+^ and allows the accommodation of larger volume changes during cycling [88]. Furthermore, the incorporation of Cu^2+^ in V_2_O_5_ led to an increase in V^4+^ content within the material. The doping process facilitated the generation of additional small polaritons by coupling the negative charge on the V^4+^ with the lattice deformation induced by the presence of Cu^2+^. As a result, the electronic conductivity of the Cu-doped V_2_O_5_ was enhanced [88]. Finally, it is also worth noting that the capacity reported is higher than that of the Y-doped V_2_O_5_ (224 mAh g^−1^) [99].

Following with Sn, Li et al. indicated that the 4% Sn^4+^-doped V_2_O_5_ presented the highest capacity (292 mAh g^−1^ and 162.5 mAh g^−1^ at 50 and 2000 mA g^−1^) and best cycling (0.36% for Sn_0.04_V_1.96_O_5−δ_ and 0.74% for V_2_O_5_ at a current density of 200 mA g^−1^) (Figure 19) [100]. The improved performance can be attributed to the presence of V^4+^ and the creation of oxygen vacancies resulting from Sn doping. DFT calculations were employed to investigate the formation of Sn-related defects and their effects. The analysis focused on point-defect formation energies and properties to determine the most probable defect and its charge compensation mechanism [101]. The study revealed that the most favorable defect is the substitution of Sn for VO, accompanied by a bound polaron at a V center serving as the charge compensation. The polaron possesses an escape barrier of 0.55 eV, enabling it to migrate away from the defect center [100]. This escape process generates charge carriers that significantly enhance the electronic conductivity compared to that of the undoped material [101]. In conclusion, the inclusion of Sn doping positively impacts electrochemical performance by providing charge carriers in the form of electron polarons and serving as nucleation sites for the formation of the lithiation phase during Li intercalation.

Studies have investigated the effects of Co doping in various lithium-based materials such as LiFePO_4_ [102,103]. In the case of LiFePO_4_, it has been observed that a high concentration of Co doping negatively impacts the performance, whereas a low level of Co doping leads to a noticeable improvement in capacity [104]. In a pioneering work by Ji et al., hydrothermally grown Co-doped V_2_O_5_ exhibited a capacity of 394.98 mAh g^−1^ after 50 cycles at 30 mA g^−1^ [105]. This improvement can be attributed to the higher electronic conductivity of the doped materials, which alleviates the strain experienced during Li^+^ insertion/extraction and slows down the structural degradation.

Cerium, being the most abundant rare earth element, possesses a relatively large radius and a strong affinity for oxygen. It is commonly selected as a structure stabilizer to enhance the ion conductivity [106]. Furthermore, its unique structure, characterized by a half-filled 4f electron orbit, allows variable electronic properties [107]. In specific, when Ce is doped into V_2_O_5_ (Ce_0.1_V_2_O_5_), microspheres with an average diameter of 2–5 μm are formed (Figure 20) and superior electrochemical performance is achieved compared to that of the undoped V_2_O_5_ [108]. The Ce-doped microspheres exhibited a high reversible capacity of 260 mAh g^−1^ at 294 mA g^−1^, with a capacity retention of approximately 88.93% after 200 cycles. This enhanced performance is attributed to the enlargement of the interplanar spacing in V_2_O_5_, which reduces the charge transfer resistance and improves the diffusion coefficient of Li^+^.

Yttrium (Y^3+^) is also an effective dopant that can significantly improve the electrochemical performance of electrode materials. This is primarily due to its large radius (90 pm), high electric charge, and strong self-polarization ability [109,110]. While there have been studies on the reversible intercalation of Y^3+^ into the host structure of V_2_O_5_, there have been no reports focusing on SIBs [111]. In a recent study, Y^3+^ was pre-intercalated into Y_0.02_V_2_O_5_, resulting in significantly enhanced cycling stability and superior rate capability. The specific capacity achieved was 119 mAh g^−1^ after 100 cycles, which is notably higher than that of the pure V_2_O_5_ electrode (84 mAh g^−1^) [95]. The improved performance of the doped V_2_O_5_ can be attributed to several factors [104]. Firstly, the Y^3+^ doping leads to an increase in low-valence vanadium ions (V^4+^), which enhances the electrical conductivity of V_2_O_5_ and facilitates the movement of charge carriers within the material. Secondly, the structural integrity of V_2_O_5_ is enhanced due to the incorporation of Y^3+^ ions maintaining the stability and integrity of the electrode material during charge and discharge cycles. Lastly, the presence of Y^3+^ ions suppresses the irreversible phase transition of hydrated V_2_O_5_, mitigating structural degradation and ensuring a more reversible electrochemical reaction. Overall, the utilization of Y^3+^ doping in V_2_O_5_ electrodes holds promise for enhancing the electrochemical performance in energy storage devices, primarily by improving electrical conductivity, enhancing structural integrity, and suppressing irreversible phase transitions [104].

The selection of the most suitable cationic dopant in V_2_O_5_ for improved electrochemical measurements and their application in energy storage systems is influenced by various factors, including the battery components and the desired performance metrics depending on the application requirements (Table 1). According to the available research data, introducing Ni into V_2_O_5_ has demonstrated enhancements in the capacity, rate capability, and cycling stability of V_2_O_5_-based electrodes. Ni doping improves charge transfer kinetics and enhances the structural stability during charge/discharge cycles. Another promising dopant is Cu, which aids in the formation of a three-dimensional nanoscale net-like structure, leading to an increased specific surface area and improved transport capacity within the electrode. While Ni and Cu have shown favorable results, other dopants such as Mn, Co, and Ce have also been explored, demonstrating positive impacts on the electrochemical performance of V_2_O_5_. Nonetheless, the optimal choice of dopant should be determined based on comprehensive studies that consider a specific battery system and performance objectives.

### 4.2. Electrolytes and Interfaces

Overcoming challenges associated with the dissolution of vanadium compounds, especially in the case of layered-type compounds, and addressing interfacial reactivity are crucial hurdles. Various strategies have been proposed to tackle these issues. An intriguing alternative involves the use of hybrid organic–water solvents in the electrolyte solution, such as the ether–water mixture. Wang et al. discovered that a hybrid electrolyte solution, specifically, tetraethylene glycol dimethyl ether (TEGDME)–water, significantly enhances the diffusion coefficient of magnesium intercalated in layered sodium vanadate (NaV_8_O_20_·nH_2_O) [112]. In this hybrid solution, TEGDME, ClO_4_^−^ ions, and water molecules coordinate with magnesium ions, as depicted in Figure 21. This coordination promotes the swift transport of cations, expanding the electrochemical stability window. The extended chain length of TEGDME strengthens interactions between water molecules (e.g., hydrogen bonding interactions) and TEGDME, consequently raising the overpotential for water splitting. The interaction between magnesium and water ligands reduces the activity of free water in the electrolyte solution, contributing to an enlarged voltage window. Furthermore, the interface layer exhibits robustness owing to the decomposition of TEGDME and the formation of a protective film. This suppresses the dissolution of vanadium, resulting in an impressive cycle life of 1000 cycles. Mechanistic studies on the electrochemical reactions confirm that Na^+^ in the original sodium vanadate undergoes deintercalation and is replaced by Mg^2+^ during cycling. The reversible intercalation of Mg^2+^ is facilitated by the shielding effect of water molecules.

Research has demonstrated that electrode materials featuring higher oxidation states of vanadium ions exhibit increased solubility in electrolyte solutions containing carbonate solvents, consequently leading to battery failure. For instance, the solubility of VO_2_F surpasses that of Li_2−x_VO_2_F. Utilizing highly concentrated solutions, such as 5.5 M LiFSI in DMC, results in a lower quantity of free solvent molecules. Consequently, this diminishes the dissolution of vanadium-based electrodes, such as Li_2−x_VO_2_F [113]. Additionally, the use of more concentrated electrolytes contributes to a reduction in interfacial resistance.

Within the hybrid aqueous/organic electrolytes, it has recently been found that the acetonitrile component of the acetonitrile–water solvent mixture suppresses the dissolution of vanadium species from layered Ca_x_V_2_O_5_·nH_2_O during electrochemical cycling, while water molecules accelerate the diffusion kinetics [76]. Another rationale for the observed enhancement upon the addition of water to the solution can be attributed to the generation of protons. Protons, formed through the dissociation of H_2_O, are incorporated into V_2_O_5_, serving as charge carriers and exhibiting a preference for binding with vanadyl oxygen. Calculations indicate that the inserted protons contribute to a reduction in the band gap and a simultaneous improvement in electronic conductivity [114]. Protons diffuse faster than magnesium in V_2_O_5_, and the preinserted protons decrease the Mg diffusion barrier.

Crafting heterostructures stands as another viable strategy for tailoring interfaces. An illustrative instance involves enhancing structural stability through bonds with carbon. Song et al. ingeniously suggested the formation of V-O-C chemical bonds to augment sodium–intercalation kinetics, resulting in an extraordinary lifespan of 3000 cycles at 5C [115]. In this context, Na_5_V_12_O_32_ is bonded to reduced graphene oxide. Computational analyses revealed a reduced energy barrier for sodium diffusion at the interface between Na_5_V_12_O_32_ and carbon.

Given that the high capacity of V_2_O_5_ has high structural instability, a promising avenue to enhance cycling stability involves integrating V_2_O_5_ with bronzes like NaV_6_O_15_ or Ca_0.17_V_2_O_5_ within a composite electrode. Consequently, the V_2_O_5_/Ca_0.17_V_2_O_5_ thin-film electrode in sodium cells exhibits reduced polarization, superior insertion kinetics, and enhanced cycling stability compared to those of standalone V_2_O_5_ [116]. It was observed that XRD reflections are displaced upon sodium intercalation, while the reflections of Ca_0.17_V_2_O_5_ are not displaced, and consequently, it is inferred that V_2_O_5_ is the host framework for sodium insertion, while Ca_0.17_V_2_O_5_ is a “pillar” for structural stability.

Intercalation of the conductive polymer poly 3,4-ethylenedioxythiophene (PEDOT) into V_2_O_5_ forms a PEDOT-V_2_O_5_ heterostructure, expansion of the interlayer spacing from 0.78 to 1.03 nm, and easier Ca intercalation [112]. Besides that, the inserted PEDOT links the V_2_O_5_ layers together, avoiding the structural collapse during the electrochemical cycling.

## 5. Conclusions

In summary, although V_2_O_5_ possesses exceptional properties for intercalation reactions, several drawbacks have prevented their implementation in more successful batteries. There is a consensus about the relevance of designing synthesis methods for controlling the microstructure and texture of the particles as a main path to optimize the electrochemistry of vanadium oxides. However, several of the synthesis methods employed in research laboratories may be not easy to scale up. On the other hand, the fabrication of nanostructures and nanoparticles with higher particle area exposed to the electrolyte solution can raise the apparent capacity to the pseudocapacitive contribution.

The mechanism of the intercalation of lithium into V_2_O_5_ is very complex and involves the formation of many phases, and this issue is typically detrimental for achieving high reversibility in rechargeable batteries. Compared to that of lithium, the mechanism for intercalation of other cations remains less explored. The sodium–vanadium bronzes, Na_x_V_2_O_5_, comprise a wide variety of structural properties. It seems that larger cations, such as potassium, tend to be more irreversibly trapped in the framework of the vanadium oxide, but we cannot completely discard that the optimization of the vanadium oxide structure and microstructure render new electrode materials for suitable potassium-ion batteries. The efficiency of intercalation of divalent cations, magnesium ions, and calcium-ions is particularly governed by the solvent properties, the co-intercalation, and the shielding of the magnesium charge. Beyond univalent and divalent cations, the intercalation of aluminum (III) potentially may be very useful to provide higher capacities. The mechanism of the aluminum-ion batteries would be strongly dependent on the electrolyte and solvent (organic, aqueous, ionic liquid and hybrid organic–aqueous), and it has still been little explored. For example, the “aluminum-intercalation” could involve the intercalation of Al^3+^, protons, AlCl_4_^−^, and other ions, depending on the electrolyte solution and the host properties. The instability of vanadium compounds in certain electrolyte solutions can be a main drawback, although this could be fixed by using a mixture of solvents. Another option to improve the electrochemistry could be simultaneously employing several ions for intercalation. Thus, the optimization of the electrolyte solutions is a main task for advancing this field. It has been found that the replacement of a small amount of vanadium by another metal can have a very positive effect in the electrochemistry, and more efforts should be addressed in this way.

In summary, the main strategies to improve the electrochemistry of the V_2_O_5_ electrode are reducing particle size, tailoring the particle morphology, modifying the interface’s electrode/electrolyte, pre-intercalation or co-intercalation, creating heterostructures, and cation or anion substitution. Having in mind the large opportunities and the systems that remain unexplored or little studied, we think that, in addition to the tremendous experimental work that remains to be done before developing futures batteries, the DFT calculations could help to pre-select the most promising materials and microstructures for post-lithium batteries.

## Figures and Tables

**Figure 2 nanomaterials-13-03149-f002:**
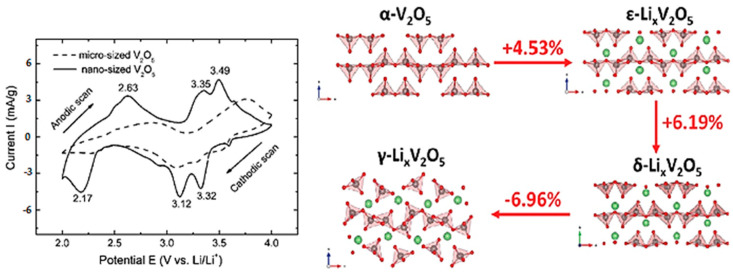
Cyclic voltammetry curves of vanadium pentoxide electrodes in lithium cell (**left**) [29]. Crystal structures of V_2_O_5_ during lithiation: α-V_2_O_5_, ε-Li_x_V_2_O_5_, δ-Li_x_V_2_O_5_, and γ-Li_x_V_2_O_5_, and the corresponding volume expansion during each phase transformation [28] (**right**). Image adapted from Refs. [28,29].

**Figure 3 nanomaterials-13-03149-f003:**
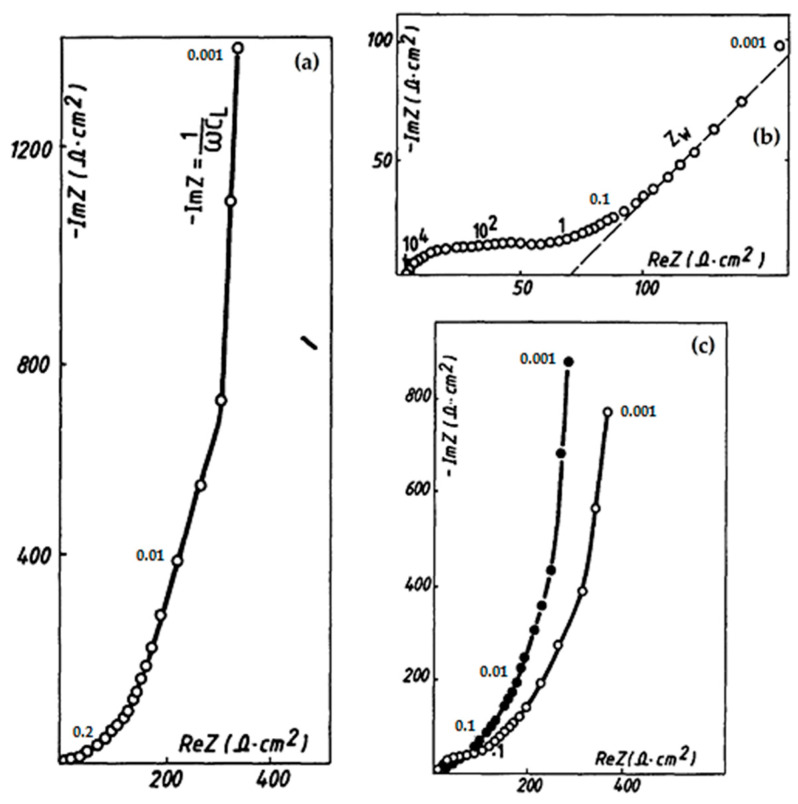
Impedance diagrams for different Li_x_V_2_O_5_ bronzes for different x values, (**a**) x = 0.01, (**b**) x = 0.2, (**c**) x = 0.36 (empty circle), and x = 0.9 (black circle) [34]. Image adapted from Ref. [34].

**Figure 4 nanomaterials-13-03149-f004:**
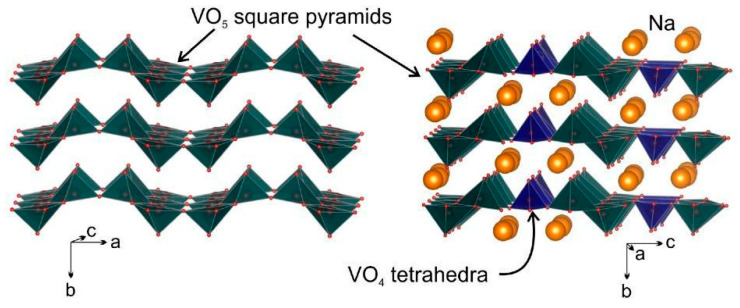
Crystal structures of α-V_2_O_5_ (**left**) and η-Na_x_V_2_O_5_ (**right**) [43].

**Figure 5 nanomaterials-13-03149-f005:**
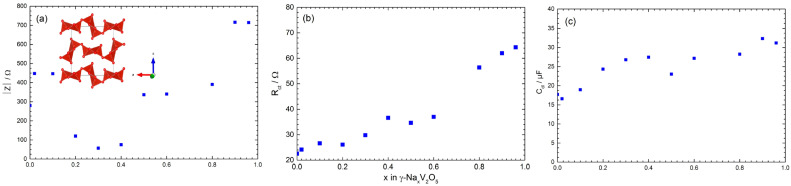
Cathode impedance and crystal structure of γ’-V_2_O_5_ as inset (**a**), charge transfer resistance R_ct_ (**b**) and double-layer capacity C_dl_ (**c**) in γ-Na_x_V_2_O_5_ [51]. Image adapted from Ref. [51].

**Figure 6 nanomaterials-13-03149-f006:**
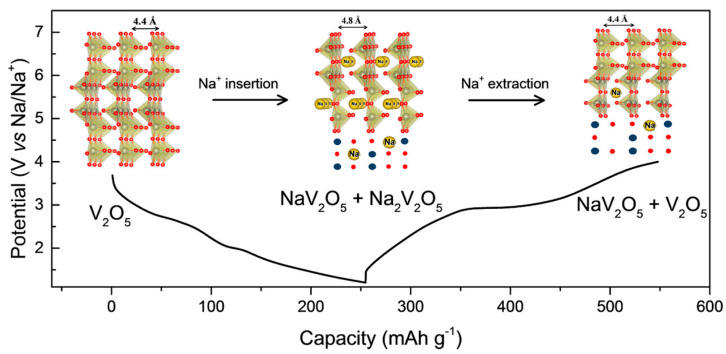
Crystal structure of V_2_O_5_ during the charge/discharge processes [54]. Image adapted from Ref. [54].

**Figure 7 nanomaterials-13-03149-f007:**
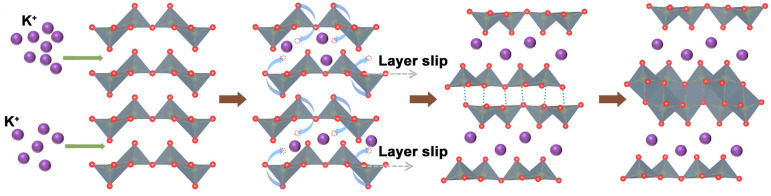
Structure of KVO [59]. Image adapted from Ref. [59].

**Figure 8 nanomaterials-13-03149-f008:**
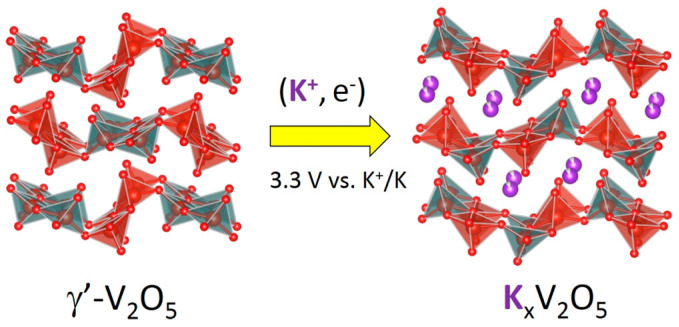
Structural transformation of γ’-V_2_O_5_ upon K^+^ ion intercalation [62]. Image adapted from Ref. [62].

**Figure 9 nanomaterials-13-03149-f009:**
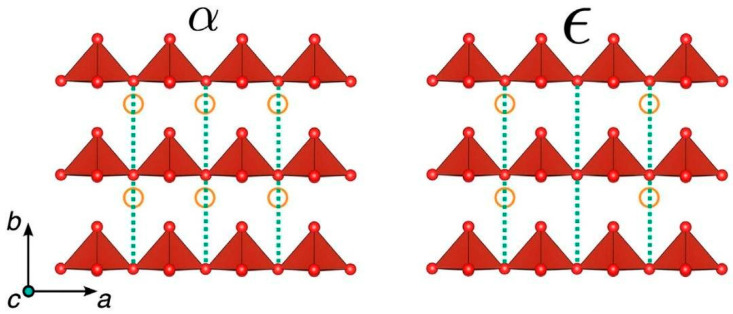
Polymorphs of V_2_O_5_ upon Mg^2+^ ion intercalation [70]. Image adapted from Ref. [70].

**Figure 10 nanomaterials-13-03149-f010:**
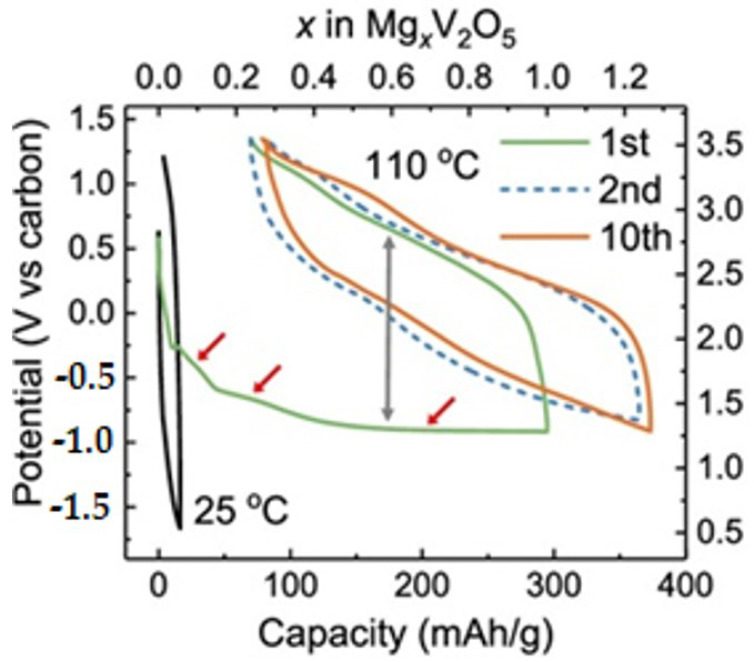
Electrochemical evaluation of Mg^2+^ intercalation into α-V_2_O_5_ in an ionic liquid Mg^2+^ electrolyte at 25 °C and 110 °C [72]. Image adapted from Ref. [72].

**Figure 11 nanomaterials-13-03149-f011:**
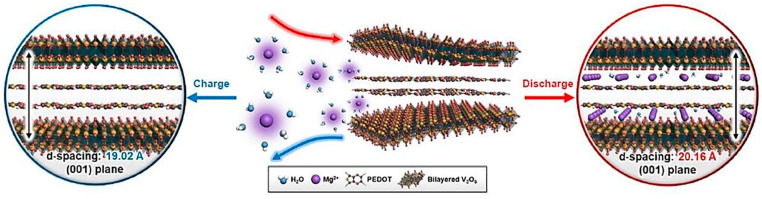
Schematic presentation of the Mg^2+^ storage mechanism in VOP [73]. Image adapted from Ref. [73].

**Figure 12 nanomaterials-13-03149-f012:**
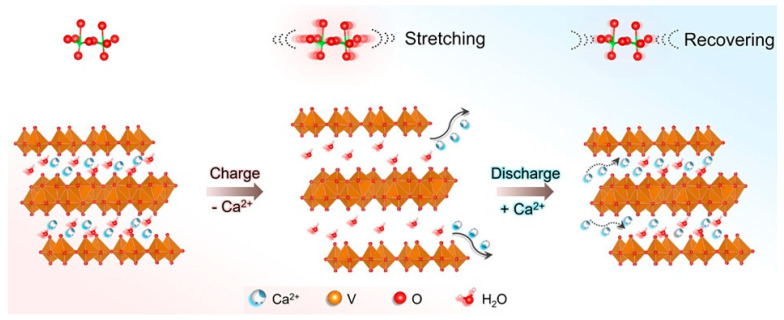
Structure evolution of V_2_O_5_ during charging/discharging processes of Ca^2+^ [76]. Image adapted from Ref. [76].

**Figure 13 nanomaterials-13-03149-f013:**
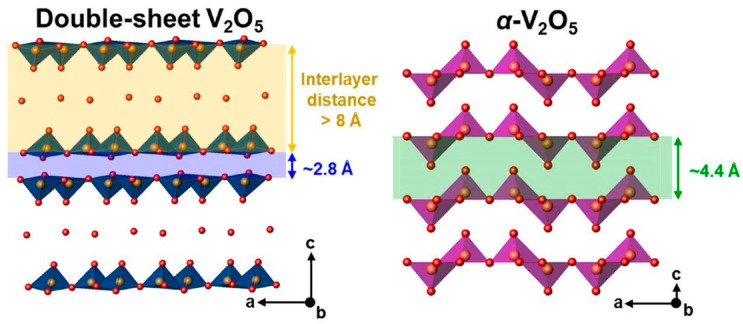
Crystal structures of double-sheet V_2_O_5_ and α-V_2_O_5_ [78]. Image adapted from Ref. [78].

**Figure 14 nanomaterials-13-03149-f014:**
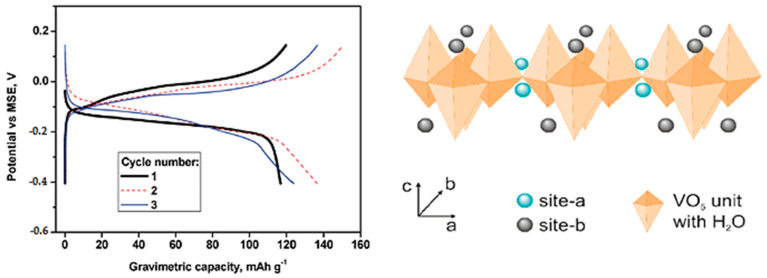
Voltage-capacity curves of V_2_O_5_ in aluminum aqueous cell (**left**). Schematic illustration of the two different intercalation sites of the VO_5_ units near the planar oxygen atom (site-a) and close to the apical oxygen atom (site-b) (**right**) [81].

**Figure 15 nanomaterials-13-03149-f015:**
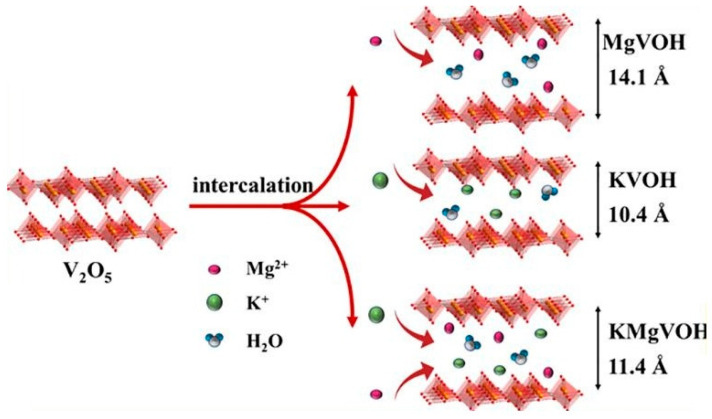
Schematic diagram of the MgVOH, KVOH, and KMgVOH crystal structures [84]. Image adapted from Ref. [84].

**Figure 16 nanomaterials-13-03149-f016:**
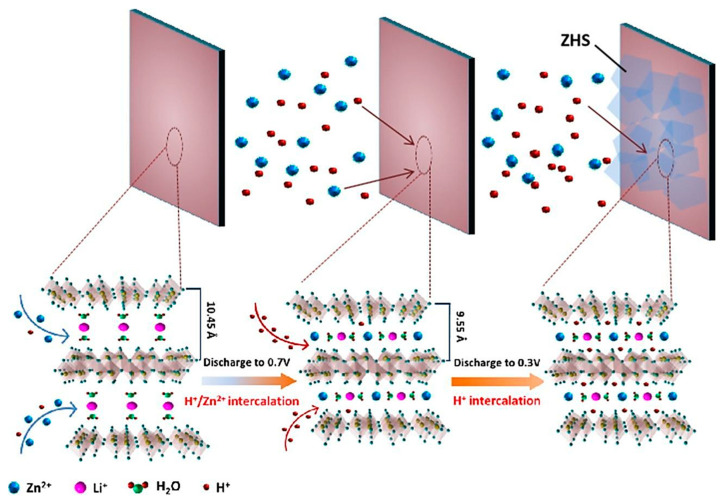
Schematic diagram of the insertion mechanism of Zn^2+^ and H^+^ [85]. Image adapted from Ref. [85].

**Figure 17 nanomaterials-13-03149-f017:**
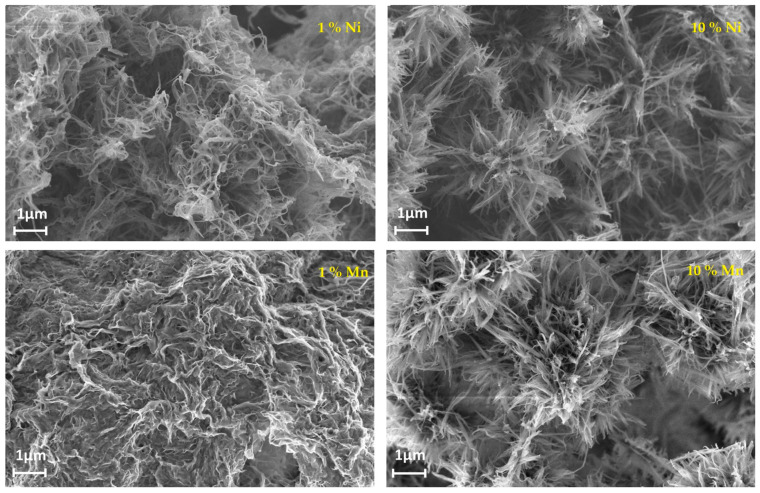
SEM images of V_2_O_5_·nH_2_O doped with 1 mol% of Ni, 10 mol% Ni, 1 mol% Mn, and 10 mol% Mn [96]. Image adapted from Ref. [92].

**Figure 18 nanomaterials-13-03149-f018:**
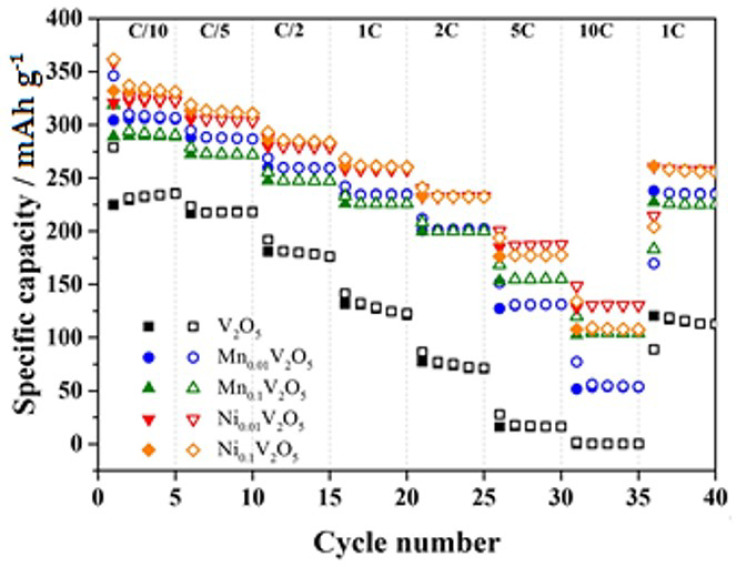
Rate test of doped and undoped sono-aerogels [96]. The cathodes, anode, and electrolyte are the doped and undoped sono-aerogels, the Li metal, and the LiClO_4_/PC, respectively. Image adapted from Ref. [92].

**Figure 19 nanomaterials-13-03149-f019:**
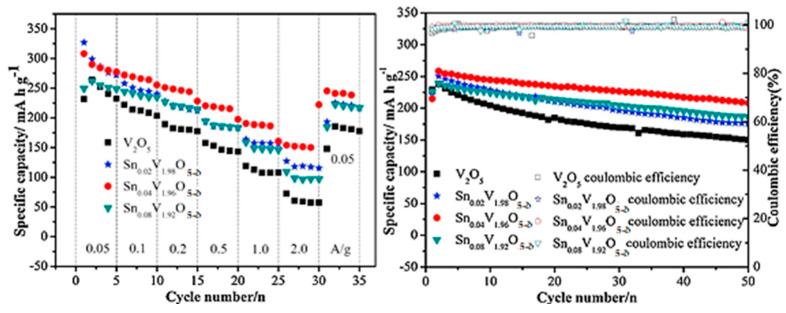
Rate tests and cycling performance of the electrodes at 200 mA g^−1^ [100]. The cathodes, anode, and electrolyte are the pure and doped vanadium pentoxide, the Li metal, and the LiPF_6_/EC/DMC, respectively. Image adapted from Ref. [100].

**Figure 20 nanomaterials-13-03149-f020:**
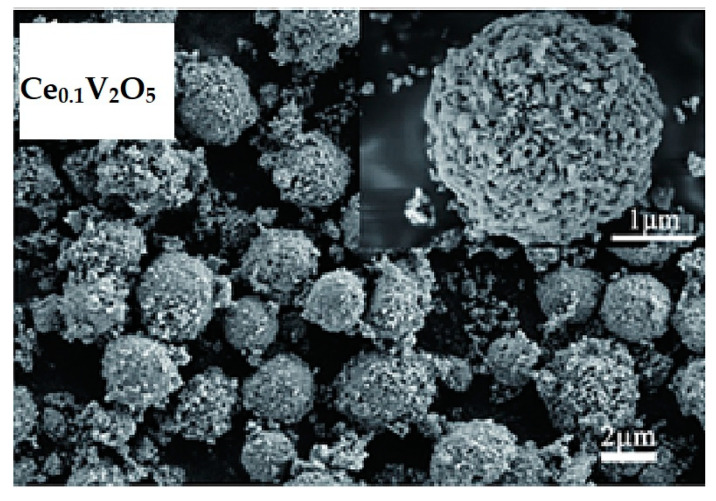
FESEM of Ce_0.1_V_2_O_5_ [106]. Image adapted from Ref. [106].

**Figure 21 nanomaterials-13-03149-f021:**
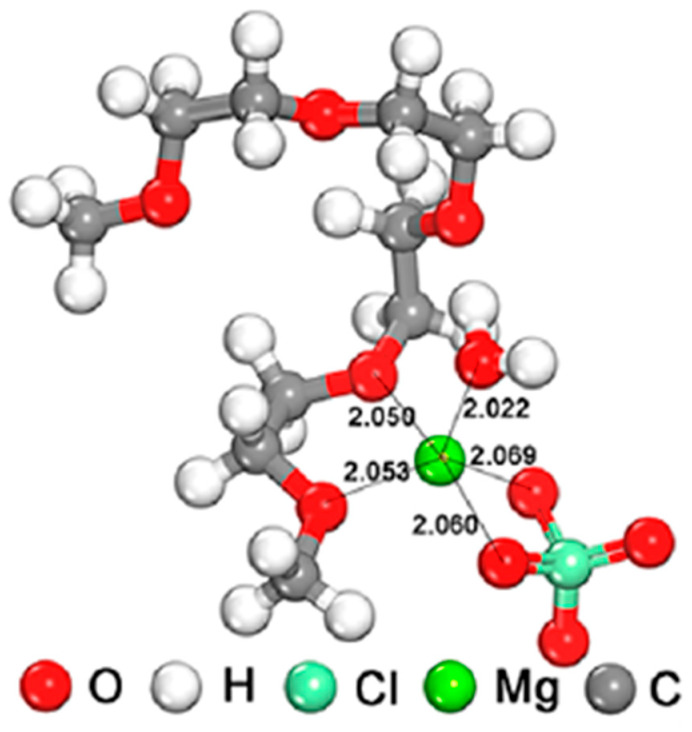
Typical solvation of Mg^2+^ in the TEGDME–water hybrid electrolyte according to Monte Carlo simulation [112]. Image adapted from Ref. [112].

**Table 1 nanomaterials-13-03149-t001:** Specific capacity and capacity retention of cationic doped V_2_O_5_.

Cathode	Specific Capacity/mAh g^−1^	Capacity Retention/%	References
Ni_0.1_V_2_O_5_ vs. Li/Li^+^	275 at 560 mA g^−1^	69.9 after 140 cycles	[92]
Mn_0.1_V_2_O_5_ vs. Li/Li^+^	225 at 560 mA g^−1^	76.4 after 140 cycles	[92]
Sn_0.04_V_1.96_O_5−δ_ vs. Li/Li^+^	292 at 200 mA g^−1^	0.36 per cycle	[96]
Co-doped V_2_O_5_ vs. Li/Li^+^	394.98 at 200 mA g^−1^	94 after 100 cycles	[105]
Ce_0.1_V_2_O_5_ vs. Li/Li^+^	260 at 294 mA g^−1^	88.93 after 200 cycles	[106]
Y_0.02_V_2_O_5_ vs. Na/Na^+^	119 at 100 mA g^−1^	0.12 per cycle between 10 and 100 cycles	[95]

## Data Availability

Data sharing is not applicable to this article.

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
