# Peer review of "Metal-Ion Intercalation Mechanisms in Vanadium Pentoxide and Its New Perspectives"

_nanomaterials, 2023, doi:10.3390/nano13243149_

Round 1
Reviewer 1 Report
Comments and Suggestions for Authors
The investigation into intercalation mechanisms in vanadium pentoxide has garnered
significant attention within the realm of research, primarily propelled by its remarkable theoretical
capacity for energy storage. This comprehensive review delves into the latest advancements that
have enriched our understanding of these intricate mechanisms. The writing is easy to understand and easy to understand。
The author has provided a systematic summary of the intercalation research on V2O5, and the overall approach is clear.
The title of this article is Intercalation mechanisms in vanadium pentoxide and its new perspectives , but the article mainly focuses on the study of metal ion intercalation. In fact, there are also many studies on the intercalation of non-metallic ions, such as the intercalation of ionic liquids.Need to supplement whether vanadium oxide has favorable competitive advantages in future development trends, or in what aspects can the performance of vanadium pentoxide be impr
Author Response
Dear Reviewer 1,
Thank you very much for your response regarding our manuscript entitled:
“Intercalation mechanisms in vanadium pentoxide and its new perspectives”
we wish to publish in Nanomaterials.We have proceeded with the revision of our manuscript and we are resubmitting our work hoping that we have fully complied with your recommendations. The highlighted revision encloses all changes in bold and underlined.
Reviewer’s comments:
The investigation into intercalation mechanisms in vanadium pentoxide has garnered significant attention within the realm of research, primarily propelled by its remarkable theoretical capacity for energy storage. This comprehensive review delves into the latest advancements that have enriched our understanding of these intricate mechanisms. The writing is easy to understand and easy to understand. The author has provided a systematic summary of the intercalation research on V2O5, and the overall approach is clear.
The title of this article is Intercalation mechanisms in vanadium pentoxide and its new perspectives, but the article mainly focuses on the study of metal ion intercalation. In fact, there are also many studies on the intercalation of non-metallic ions, such as the intercalation of ionic liquids. Need to supplement whether vanadium oxide has favorable competitive advantages in future development trends, or in what aspects can the performance of vanadium pentoxide be improved.
Response to reviewer: Thank you very much for the comment. The title has been changed, accompanied by noteworthy revisions in both the text and figures.
With our best regards,
Ricardo Alcántara and Dimitra Vernardou

Reviewer 2 Report
Comments and Suggestions for Authors
This manuscript provides overall information related to intercalation mechanisms in vanadium pentoxide materials. This review paper provides the latest advancements in the materials including exceptional capacity, intrinsic mechanisms, and structural properties along with perspectives. As the vanadium pentoxide can be adopted not only currently used energy storage systems such as lithium ion battery, but it is further expanded to next-generation systems including sodium, potassium, magnesium, aluminum etc., this review paper is acceptable in this journal after minor revision.
(1) If the journal accept table for navigating the content of this review paper, it would be recommend add the table in the initial part of the manuscript for authors to more easily access the wanted content.
(2) In part 3, authors provide intercalation chemistry of diverse cations in the vanadium pentoxide material. Like Figure 5, it is better to provide theoretical or experimental potential vs. capacity profiles to all intercalation chemistries introduced in this paper.
Author Response
Dear Reviewer 2,
Thank you very much for your response regarding our manuscript entitled:
“Intercalation mechanisms in vanadium pentoxide and its new perspectives”
we wish to publish in Nanomaterials. We have proceeded with the revision of our manuscript and we are resubmitting our work hoping that we have fully complied with your recommendations. The highlighted revision encloses all changes in bold and underlined.
Reviewer’s comments:
This manuscript provides overall information related to intercalation mechanisms in vanadium pentoxide materials. This review paper provides the latest advancements in the materials including exceptional capacity, intrinsic mechanisms, and structural properties along with perspectives. As the vanadium pentoxide can be adopted not only currently used energy storage systems such as lithium ion battery, but it is further expanded to next-generation systems including sodium, potassium, magnesium, aluminum etc., this review paper is acceptable in this journal after minor revision.
(1) If the journal accept table for navigating the content of this review paper, it would recommend add the table in the initial part of the manuscript for authors to more easily access the wanted content.
Response to reviewer: Thank you very much for the comment. I believe this is a matter of the Journal’s requirements.
(2) In part 3, authors provide intercalation chemistry of diverse cations in the vanadium pentoxide material. Like Figure 5, it is better to provide theoretical or experimental potential vs. capacity profiles to all intercalation chemistries introduced in this paper.
Response to reviewer: Thank you very much for the comment. We agree with the reviewer that these changes can render the manuscript more useful for the potential readers. Cyclic voltammetry curves have been added to Figure 2 along with voltage-capacity curves in Figure 14.
With our best regards,
Ricardo Alcántara and Dimitra Vernardou

Reviewer 3 Report
Comments and Suggestions for Authors
Vanadium pentoxide is a highly versatile material, easily tunable by controlling the synthesis parameters, which can serve many applications in energy storage; for sure this topic deserves a review article, of great interest for Nanomaterials journal. Alcántara et al. describe the mail results obtained about the use of V2O5 in many different batteries, based on both univalent and multivalent ions. However, some aspects should be improved before the manuscript will be suitable for publication.
In the introduction, the Authors say that the presence of protons corresponds to the reduction of certain vanadium ions from V5+ to V4+. I don’t think it is a redox reaction (in that case, which species is oxidized?), but rather an acid-base equilibrium.
The section about V2O5 synthesis should be expanded: which are the “chemical reagents” more commonly used? And the chelating agent? I would suggest to introduce a table to summarize which are the most commonly used precursors, the adopted conditions and the properties of the produced material in terms, for instance, of specific surface area, porosity and particle size (when available).
Describing the electrospinning, is the polymer completely converted into carbon upon calcination or is it partially destroyed? The remaining fibers, are they fibers of V2O5 or fibers of C? If present, a TEM image would be useful.
Concerning lithium intercalation, at which electrochemical potentials the listed reactions take place (lines 228-230)? Are there some thermodynamic data about the phase transitions depicted in figure 2?
Li insertion was studied experimentally by Farcy et al. by using EIS, could the Authors report some graphs and comment them more in detail? In order to deepen a little bit the discussion.
I would recommend to merge figures 3 and 4. Is the material undergoing all the phases upon Na insertion? Is the sequence in the pictures alpha, gamma and eta? Could the Authors comment the transformation from one phase to the other?
About Na intercalation/deintercalation, the Authors mentioned many characterization techniques. I would suggest to reproduce some figures from the literature, in order to show the most significant results and help the reader to follow the process.
Figure 9 is not clear without a proper discussion: it is oversimplified, likely it was used as graphical abstract, but to be used in a review article it should be adjusted and properly explained.
The comparison between the two Ca-intercalated structures proposed in figures 11 and 12 is interesting, but not self-evident from the two pictures (also because of the different perspectives and color codes). I would suggest to make a unified figure, with a uniform style, in order to clearly pointing out the differences between the two proposed models.
The mechanism of intercalation of Al3+ varies depending on the amount of x·e- that are transferred. Are there some graphs in the literature of capacity vs potential upon galvanostatic cycling? In this way, it would be more evident the electrochemical behavior.
In figures 18 and 19, the scheme of LIBs should be explicitly defined in the labels (cathode|electrolyte|anode).
In Table 1 some current densities are reported as C/n, but this number refers to a theoretical capacity that can be considered differently by the different scholars. Therefore, I recommend to check the theoretical capacity and the active mass of the electrode in the cited papers, in order to extrapolate the absolute current density in mA/g (so that all the values of the table can be compared in the same conditions).
I am sure that after fixing these issues, the scientific soundness of the review article will considerably increase, becoming a benchmark for the use of V2O5 in energy storage devices.
Comments on the Quality of English Language“Oxonium” cannot stay alone… “oxonium cations” is the proper definition.
I would not speak about “hydrated protons” (of course protons in water form hydroxonium cations, but usually in the chemical textbooks they are just referred to as “protons”).
“Jacobsen was first conducted”… “was” should be deleted.
Author Response
Dear Reviewer 3,
Thank you very much for your response regarding our manuscript entitled:
“Intercalation mechanisms in vanadium pentoxide and its new perspectives”
we wish to publish in Nanomaterials.We have proceeded with the revision of our manuscript and we are resubmitting our work hoping that we have fully complied with your recommendations. The highlighted revision encloses all changes in bold and underlined.
Reviewer’s comments:
Vanadium pentoxide is a highly versatile material, easily tunable by controlling the synthesis parameters, which can serve many applications in energy storage; for sure this topic deserves a review article, of great interest for Nanomaterials journal. Alcántara et al. describe the main results obtained about the use of V2O5 in many different batteries, based on both univalent and multivalent ions. However, some aspects should be improved before the manuscript will be suitable for publication.
Thank you very much for your comments/suggestions. Please see below our response.
1) In the introduction, the Authors say that the presence of protons corresponds to the reduction of certain vanadium ions from V5+ to V4+. I don’t think it is a redox reaction (in that case, which species is oxidized?), but rather an acid-base equilibrium.
Response to reviewer: Thank you very much for the comment. The text has been modified accordingly.
2) The section about V2O5 synthesis should be expanded: which are the “chemical reagents” more commonly used? And the chelating agent? I would suggest to introduce a table to summarize which are the most commonly used precursors, the adopted conditions and the properties of the produced material in terms, for instance, of specific surface area, porosity and particle size (when available).
Describing the electrospinning, is the polymer completely converted into carbon upon calcination or is it partially destroyed? The remaining fibers, are they fibers of V2O5 or fibers of C? If present, a TEM image would be useful.
Response to reviewer: Thank you very much for the comment. The text has been modified accordingly.
3) Concerning lithium intercalation, at which electrochemical potentials the listed reactions take place (lines 228-230)? Are there some thermodynamic data about the phase transitions depicted in figure 2?
Response to reviewer: Thank you very much for the comment. The electrochemical potentials are indicated in the text for Li0.5V2O5, LiV2O5 and Li2V2O5.
There is no thermodynamic data reported in Ref. [28] for the α-, ε-, γ- and δ-V2O5.
4) Li insertion was studied experimentally by Farcy et al. by using EIS, could the Authors report some graphs and comment them more in detail? In order to deepen a little bit, the discussion.
Response to reviewer: Thank you very much for the comment. The text has been modified accordingly including the addition of a Figure.
5) I would recommend to merge figures 3 and 4. Is the material undergoing all the phases upon Na insertion? Is the sequence in the pictures alpha, gamma and eta? Could the Authors comment the transformation from one phase to the other?
Response to reviewer: Thank you very much for the comment.Τhe material reported in Ref. [41] is only correlated with η-NaxV2O5. In the manuscript, you can find information regarding each phase.
We believe that Figures 3 and 4 cannot be merged because Figure 4 refers to a metastable phase. The text has been modified accordingly to make this part clearer.
6) About Na intercalation/deintercalation, the Authors mentioned many characterization techniques. I would suggest to reproduce some figures from the literature, in order to show the most significant results and help the reader to follow the process.
Response to reviewer: Thank you very much for the comment. The text has been modified accordingly with the addition of Figures to show the most significant results.
7) Figure 9 is not clear without a proper discussion: it is oversimplified, likely it was used as graphical abstract, but to be used in a review article it should be adjusted and properly explained.
Response to reviewer: Thank you very much for the comment. The text and Figure have been modified accordingly.
8) The comparison between the two Ca-intercalated structures proposed in figures 11 and 12 is interesting, but not self-evident from the two pictures (also because of the different perspectives and color codes). I would suggest to make a unified figure, with a uniform style, in order to clearly pointing out the differences between the two proposed models.
Response to reviewer: Thank you very much for the comment. Figure 11 has been removed to avoid misunderstanding and Figure 12 has been replaced with the structure evolution of V2O5 during charging/discharging processes.
9) The mechanism of intercalation of Al3+ varies depending on the amount of x·e- that are transferred. Are there some graphs in the literature of capacity vs potential upon galvanostatic cycling? In this way, it would be more evident the electrochemical behavior.
Response to reviewer: Thank you very much for the comment. A corresponding graph has been included.
10) In figures 18 and 19, the scheme of LIBs should be explicitly defined in the labels (cathode|electrolyte|anode).
Response to reviewer: Thank you very much for the comment. The information related with the cathode, electrolyte and anode in both Figures 18 and 19 has been included in the revised manuscript.
11) In Table 1 some current densities are reported as C/n, but this number refers to a theoretical capacity that can be considered differently by the different scholars. Therefore, I recommend to check the theoretical capacity and the active mass of the electrode in the cited papers, in order to extrapolate the absolute current density in mA/g (so that all the values of the table can be compared in the same conditions).
Response to reviewer: Thank you very much for the comment. The values have been changed according to your suggestion.
12) Comments on the Quality of English Language
“Oxonium” cannot stay alone… “oxonium cations” is the proper definition.
I would not speak about “hydrated protons” (of course protons in water form hydroxonium cations, but usually in the chemical textbooks they are just referred to as “protons”).
“Jacobsen was first conducted”… “was” should be deleted.
Response to reviewer: Thank you very much for the comment. The corresponding changes have been performed.
With our best regards,
Ricardo Alcántara and Dimitra Vernardou

Round 2
Reviewer 3 Report
Comments and Suggestions for Authors
The Authors have carefully replied to all the comments, considerably improving the completeness and clarity of the manuscript. I really appreciate their efforts and I think that the manuscript is now suitable for publication on Nanomaterials journal.